# Mastering Symbolic Operations: Augmenting Language Models with Compiled Neural Networks

**Yixuan Weng** [1*], **Minjun Zhu**[1,2]*, **Fei Xia**[1,2], **Bin Li**[3], **Shizhu He**[1,2,✉] , **Kang Liu**[1,2,✉], **Jun Zhao**[1,2]

[1] The Laboratory of Cognition and Decision Intelligence for Complex Systems, IA, CAS
[2] School of Artificial Intelligence, University of Chinese Academy of Sciences
[3] College of Electrical and Information Engineering, Hunan University
wengsyx@gmail.com, {shizhu.he, kliu, jzhao}@nlpr.ia.ac.cn

## Abstract

Language models' (LMs) proficiency in handling deterministic symbolic reasoning and rule-based tasks remains limited due to their dependency implicit learning on textual data. To endow LMs with genuine rule comprehension abilities, we propose "Neural Comprehension" - a framework that synergistically integrates compiled neural networks (CoNNs) into the standard transformer architecture. CoNNs are neural modules designed to explicitly encode rules through artificially generated attention weights. By incorporating CoNN modules, the Neural Comprehension framework enables LMs to accurately and robustly execute rule-intensive symbolic tasks. Extensive experiments demonstrate the superiority of our approach over existing techniques in terms of length generalization, efficiency, and interpretability for symbolic operations. Furthermore, it can be applied to LMs across different model scales, outperforming tool-calling methods in arithmetic reasoning tasks while maintaining superior inference efficiency. Our work highlights the potential of seamlessly unifying explicit rule learning via CoNNs and implicit pattern learning in LMs, paving the way for true symbolic comprehension capabilities. The code is released at: https://github.com/wengsyx/Neural-Comprehension.

## 1 Introduction

Language models (LMs), particularly large language models (LLMs), have exhibited impressive performance on complex reasoning tasks (Brown et al., 2020; Zhang et al., 2022; Chowdhery et al., 2022; Wei et al., 2022a; Suzgun et al., 2022). Despite this, the proficiency of LMs in tackling deterministic symbolic reasoning and rule-based tasks is still limited (Welleck et al.; Razeghi et al., 2022). For example, GPT-3's arithmetic performance declines with higher digit numbers (Brown et al., 2020), and its mathematical accuracy is influenced by word frequency in training data (Razeghi et al., 2022). Moreover, length

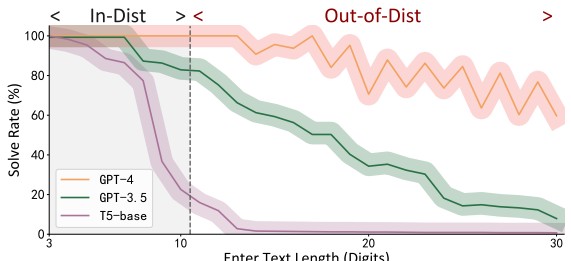

Figure 1: The length generalization of T5 (with fine-tune), GPT-3.5 and GPT-4 (with few-shot) on symbolic operations (Addition) tasks. To evaluate the model's proficiency, we conducted experiments on tasks ranging from 3 to 30 digits, with longer than 10 digits being out-of-distribution of training data.

generalization (Anil et al., 2022) remains a challenge even for 100-billion-parameter models, such as GPT-4 (Bubeck et al., 2023). We hypothesize that these limitations stem from the dependency of LMs on implicitly learning rules from textual data. As illustrated in Figure 1, a simple length generalization experiment using addition tasks with varying numbers of digits highlights this limitation. Performance deteriorates as test length increases, indicating that these models strongly rely on statistical patterns in the data rather than capturing fundamental logical structures. This implicit learning of statistical patterns constrains the accuracy of LMs in executing symbolic operations tasks.

---

*These authors contribute equally to this work. And ✉ means corresponding author.

We propose a transformer-based language model framework, termed "Neural Comprehension", which synergistically integrates a pretrained LM (Li et al., 2021b) and compiled neural networks (CoNNs) (Weiss et al., 2021), combines their complementary strengths in a plug-and-play manner, to achieve high accuracy and robust performance. CoNNs are neural networks but the rules are explicitly coded through transformer-liked structures and attention. Therefore, the CoNN is human-controllable, executing rules through artificially generated attention weights, and can achieve perfect accuracy once compiled network is done. Neural Comprehension relys solely on neural networks without requiring additional tools. It employs a token-by-token generative method, analogous to GPT-3, where each token can be generated by either the pretrained LM or one of the CoNNs. The Neural Comprehension comprises a pretrained LM and multiple sets of CoNNs. The implementation of the Neural Comprehension framework facilitates the integration of rule-intensive abilities and reasoning capabilities into LMs, endowing them with genuine symbolic comprehension skills.

We conduct extensive experiments to evaluate the performance of our proposed Neural Comprehension method on a variety of rule-intensive tasks. Our experimental results demonstrate the effectiveness of our approach in comparison with existing state-of-the-art techniques, such as vanilla fine-tuning, few-shot learning, and Chain-of-Thought reasoning (Wei et al., 2022b). Specifically, Neural Comprehension outperforms these methods in terms of accuracy, efficiency, and interpretability, showcasing its superiority in handling rule-intensive tasks. On the other hand, compared to the Tool-Based method (Mialon et al., 2023), Neural Comprehension provides a unified end-to-end neural network framework, eliminating the need for external interpreters and having higher inference efficiency. Historically, LMs are far from mastering robust symbolic task, such as symbolic operations and arithmetic reasoning (Stolfo et al., 2023). Our research provides a compelling case for LMs in neural network frameworks mastering symbolic operations, highlighting its potential to transform the landscape of symbolic reasoning and numerical computation capabilities for LMs.

**Contributions**   Our main contributions are as follows:

- We pioneer the implementation of flawless execution rule-intensive symbolic operations for language models that rely on neural networks. By employing a plug-and-play way, we successfully integrate CoNNs, which are interpretable and human-controllable, into the language model. Our method facilitates direct rule deduction without the need for learning from conditional probabilities, leading to a more robust and effective approach. (**Section** 4)

- We have built the AutoCoNN toolkit to make Neural Comprehension scalable, which leverages LLMs' contextual learning capabilities to automatically generate new CoNNs. Our method can be easily extended to various symbolic operations tasks. (**Seciton** 4.3)

- Our experimental results on symbolic tasks and real-world arithmetic reasoning tasks demonstrate the superior performance of our method in comparison to existing techniques. Notably, our LM achieves flawless execution on symbolic reasoning tasks. (**Section** 5.1 5.2)

- It is worth noting that tool-based methods are only applicable to language models with code generation capabilities and require the cooperation of external interpreters. Our experiments demonstrate that the symbolic processing capabilities of neural understanding are on par with tool-based methods, but are applicable to models ranging from small ones like T5-Small (60M) to large ones like GLM-130B (130B) and GPT-3.5 (175B). (**Section** 5.3)

- We also studied the potential of combining multiple CoNNs and found that adding correlated CoNNs can continuously increase performance, while adding uncorrelated CoNNs rarely leads to performance degradation. This provides a new approach for model fusion, enabling the model to easily acquire new knowledge. (**Section** 5.4)

## 2   RELATED WORKS

**Pretrained Language Models** encompass those trained on general-purpose corpora (Lewis et al., 2019; Scao et al., 2022; Sun et al., 2022) and specialized symbolic tasks (Geva et al., 2020; Lewkowycz et al., 2022; Yang et al., 2023). They primarily aim to capture statistical patterns in language, which limits their capacity for symbolic reasoning. Symbolic reasoning involves manipulating abstract symbols and logical rules to derive new knowledge (Shindo et al., 2021; Yang & Deng, 2021) and necessitates the ability to extrapolate to novel situations and reason about concepts

absent in the training data (Fujisawa & Kanai, 2022). Due to the constraints of gradient learning, LMs face challenges in wholly solving symbolic problems (Stolfo et al., 2023).

**In-Context Learning** has emerged as a promising approach to address these challenges (Dong et al., 2022) and closely approximate the predictors computed by gradient descent (Akyürek et al., 2022). By prompting the language model to generate an explanation before generating an answer, the chain of thought (Wei et al., 2022b; Kojima et al., 2022; Zhang et al., 2023; Zhou et al., 2022a) encourages the model to think sequentially. This technique has been employed in various numerical and symbolic reasoning tasks, such as scratchpad prompting (Nye et al., 2021) for length generalization (Anil et al., 2022) and utilizing the chain of thought to perform arithmetic operations like summing pairs of single digits with carry (Zhou et al., 2022b). However, this approach often necessitates substantial computational resources, and achieving perfect accuracy remains challenging.

**Augmented Language Models** have been proposed as an alternative, supplementing language models with external tools (Mialon et al., 2023). Examples include generating Python code for numerical reasoning (Gao et al., 2022; Chen et al., 2022) or incorporating tool usage as a pretraining task (Schick et al., 2023). However, using external tools lacks a unified framework with language models and instead relies on the normativity of program generation.

## 3 PRELIMINARIES

---

**Example 1  The Parity CoNN**

If we want the transformer to perform the Parity task full accurately:
   **Input:**   1   0

1. **Select**(**Indices**,**Indices**,**True**) = $\begin{bmatrix} 1 & 1 \\ 1 & 1 \end{bmatrix}$

2. **Aggregate**$\left( \begin{bmatrix} 1 & 1 \\ 1 & 1 \end{bmatrix}, \begin{bmatrix} 1 & 0 \end{bmatrix} \right) = \begin{bmatrix} 1 & 1 \end{bmatrix}$

3. **Zipmap**($\begin{bmatrix} 1 & 1 \end{bmatrix}$, **Lambda**   $x : 0$ **if** $x \% 2 == 0$ **else** $1$) = $\begin{bmatrix} 1 & 1 \end{bmatrix}$

   **Output:**   $\begin{bmatrix} 1 & 1 \end{bmatrix}$

---

Figure 2: Demonstration of the principles of `Parity` CoNN.

**Compiled Neural Network (CoNN).** CoNN is a transformer-based neural network leveraging artificially compiled attention weights to execute rules. CoNN has multiple attention layers and Multi-Layer Perceptron (MLP) layers, and each attention layer facilitates interactions between tokens. For example, in Figure 2, the multiplication of query and key elements representing a "**Select**" operation in CoNN. Subsequent multiplication with value elements indicates an "**Aggregate**" operation. The MLP layer is responsible for the token itself and is referred to as the "**Zipmap**" operation (Weiss et al., 2021). By utilizing the three operations (Select, Aggregate, and Zipmap) to represent the sequence-to-sequence process, we can convert symbolic parity task into transformer weights (Lindner et al., 2023). CoNN can also stack multiple layers to address various human-defined rule problems, such as mathematical calculations and symbol operations [1]. Figure 2 shows an example of `Parity` CoNN: The first step is to obtain a matrix of all ones to that of the sequence using the "**Select**" operation. Secondly, the "**Aggregate**" operation is used to combine the matrix obtained in the previous step with the input sequence (with the aim of calculating the total number of 0's and 1's in the sequence). The third step involves determining whether the total count is odd or even by "**Zipmap**".

## 4 METHOD

Language models excel in language understanding tasks, but lack robust capabilities for symbolic tasks. We propose a Neural Comprehension framework that make CoNN guided by abstract rules into language models in a plug-and-play fashion, which integrates the language model's implicit learning parameters and CoNNs' explicit learning parameters. In Neural Comprehension, we designed CoNNs in neural networks with weights and structures directly encoding the symbolic rules within the standard architecture of the LM to enhance deterministic rule comprehension and allow for deterministic execution of rule-based tasks.

---

[1] **Appendix B** provides a more detailed description of CoNN.

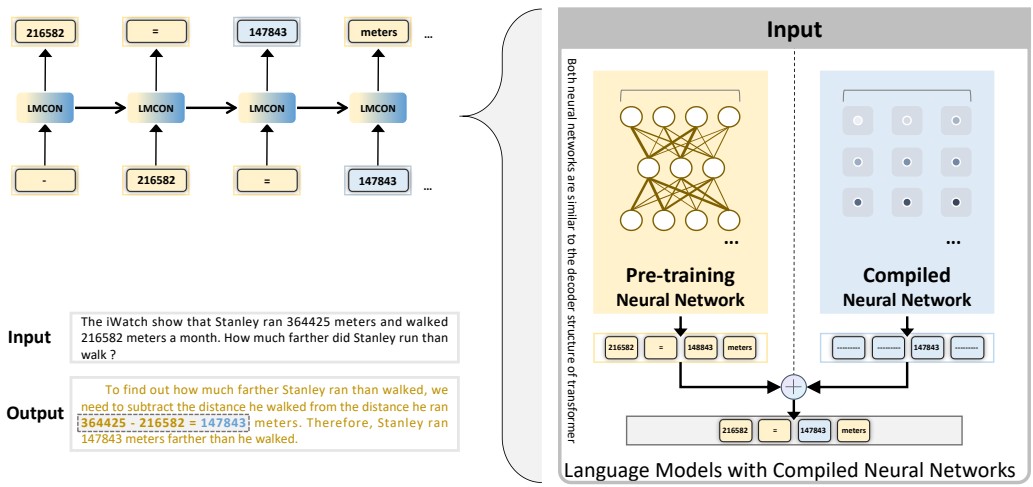

Figure 3: The architecture of the proposed Neural Comprehension framework.

## 4.1 NEURAL COMPREHENSION

Neural Comprehension is a MoE-style (Shazeer et al., 2017) neural network framework we designed, which is entirely composed of neural networks and maintains the generative seq2seq style. It uses predefined CoNNs as gating to determine when to output results from CoNNs. This approach is simple and plug-and-play, allowing combination with pretrained LMs. As illustrated in Figure 3, the language model encodes the context and produces the textual and reasoning process context $D(x)$ step by step, a decoder architecture to generate the subsequent context step by step iteratively. And CoNNs handle sequence transformations involving rules, when a rule-required operation emerges, CoNN's attention is utilized to calculate specific values. For example, in Figure 3, when calculating *364425-216582*, the only pretrained language model may output *148843*, which is incorrect. However, the `Subtraction` CoNN can correct the result to *147843* in the neural comprehension framework. This dynamically encoded process improves intermediate results interpretability and final result accuracy. Neural Comprehension combines LM and CoNNs in a piecewise function to perform gradient update. LM's hidden state output is $H_L = \left( H_{L_1} \cdots H_{L_{d_L}} \right)^\top \in \mathbb{R}^{d_L}, \quad H_{L_i} \in (0, 1)$, and CoNN's output is $H_C = \left( H_{C_1} \cdots H_{C_{d_C}} \right)^\top \in \mathbb{R}^{d_C}, \quad H_{C_i} \in \{0, 1\}$,

$$\hat{i} = \underset{i}{\mathrm{argmax}} \left[ \begin{pmatrix} I_{d_L}, 0 \\ 0, \beta I_{d_C} \end{pmatrix} \begin{pmatrix} H_L, \\ H_C, \end{pmatrix} \right], \quad \beta \in \{0, 1\} \tag{1}$$

where $H_C$ is a one-hot vector, and $d_L$ and $d_C$ here refer to the vocabulary size of the Model's decode output. [2] The Neural Comprehension combines the LLM's hidden state output, $H_L$, and CoNN's output, $H_C$, using identity matrices $I_{d_L}$ (for $d_L$) and $I_{d_C}$ (for $d_C$) to concatenate them for model fusion. Specifically, the hidden state representation matrix is obtained by extending the original hidden state representation matrix of the LM with the hidden state matrix on the CoNNs' vocabulary through a block matrix operation, resulting in a larger matrix.

## 4.2 GRADIENT MODIFICATION IN NEURAL COMPREHENSION

To better appreciate the benefits of our method in handling rule-intensive tasks and improving accuracy, it is crucial to understand the gradient perspective of In-Context Learning (ICL). Recent studies on ICL algorithms have shown that the learning process of language models within the optimization process in ICL can be viewed as a search for suitable gradients to minimize the loss

---

[2]It is worth noting that in this paper, for ease of implementation, the output vocabulary size of CoNNs' decode $d_C$ is generally less than 100 due to limitations in computing resources (detailed information is shown in **Appendix Table 1**).

function. (Garg et al., 2022; Von Oswald et al., 2023). Due to the implicit learning nature of standard ICL methods, gradients learned from data may not always be ideal for addressing rule-intensive tasks. Therefore, our proposed method introduces an explicit learning component to provide more appropriate gradient updates for such tasks, ultimately leading to enhanced overall performance. We focus on elucidating the changes in the gradient introduced by the Neural Comprehension model, the gradient of the model during the execution of ICL can be partitioned into two categories based on the origin of the gradients:

$$\text{Gradient} = \begin{cases} G_T & \text{Text: Language Model} \\ G_R & \text{Rule: CoNNs} \end{cases} \tag{2}$$

Here, $G_T$ represents the gradients derived implicitly from the language model (LM) and corresponds to the text-based learning aspect of the model. Conversely, $G_R$ represents the gradients explicitly derived from the CoNNs, encoding rule-based knowledge. The specific computation process can be seen in Equation 1. Note that the gradients' decomposition is only approximate and may not reflect the true generating process of text. The Neural Comprehension framework integrates both gradient sources to optimize the ICL process. In linear regression problems (Akyürek et al., 2022), the loss function can be expressed as a piecewise function according to Equation 1, here $P_1(x)$ is the LM and $P_2(x)$ is CoNN, the In-context-learner can be separate into two process:

$$L = \left\| y - \beta^\top x \right\|^2 \tag{3}$$

$$= \begin{cases} \left\| y - G_T^\top x \right\|^2 & x \in P_1(x) \\ \left\| y - G_R^\top x \right\|^2 & x \in P_2(x) \end{cases} \tag{4}$$

Based on the partitioned gradient as defined in Equation 2, the overall gradient of the Neural Comprehension model can be obtained by computing their individual gradients concerning the respective $\beta$:

$$\underbrace{\frac{\partial L}{\partial \beta}}_{\text{Gradient}} = \begin{cases} \frac{\partial L}{\partial G_T} & x \in P_1(x) \\ \frac{\partial L}{\partial G_R} & x \in P_2(x) \end{cases} \tag{5}$$

This partitioning allows the Neural Comprehension model to specifically address the gradient requirements of both implicit learning via LM and explicit learning via CoNNs. It is crucial to note that CoNNs are designed to minimize the loss associated with rule-based tasks, essentially providing an optimal gradient for tasks involving rule-intensive operations. This leads to a substantial improvement in the model's accuracy for rule-based tasks, as the gradient updates provided by CoNNs are more suitable for rule learning compared to the initially available gradients from the LM. By amalgamating both of gradient sources, the Neural Comprehension model achieves a more refined optimization of in-context learning. The Neural Comprehension model effectively balances the need for implicit and explicit learning within the ICL framework, leading to enhanced overall performance in terms of accuracy and interpretability.

## 4.3 AUTOCONN TOOLKIT

To improve the scalability of Neural Comprehension frameworks, we propose the AutoCoNN toolkit, which can automatically generate new CoNNs and adapt Neural Comprehension given by "Instruct" and "Example", where "Instruct" serves as explicit symbolic definitions, and "Example" provides some input-output pairs for the operation. Considering that LLMs like GPT-4 can describe symbolic reasoning processes but cannot faithfully execute them (Cai et al., 2023), AutoCoNN Toolkit enables converting the reasoning process of symbols into CoNNs while maintaining a complete neural network framework without extra interpreters. The AutoCoNN process is divided into three steps. First, we provide 24 Tracr code (Lindner et al., 2023) examples as context. Then the LLM is asked to generate 20 different Tracr codes by sampling decoding based on the "Instruct" and "Example". Finally, we convert these codes into pytorch-form (Paszke et al., 2019) CoNNs and filter them on the pre-provided Examples to obtain accurate CoNNs. We provide further experimental analysis to test the performance of AutoCoNN in Appendix D. Meanwhile, the `Parity`, `Reverse`, `Copy` and `Last Letter` CoNN mentioned in the **Section** 5 are constructed by AutoCoNN, which demonstrates the practical value of AutoCoNN.

# 5    EXPERIMENTS AND RESULT

## 5.1    SYMBOLIC TASKS

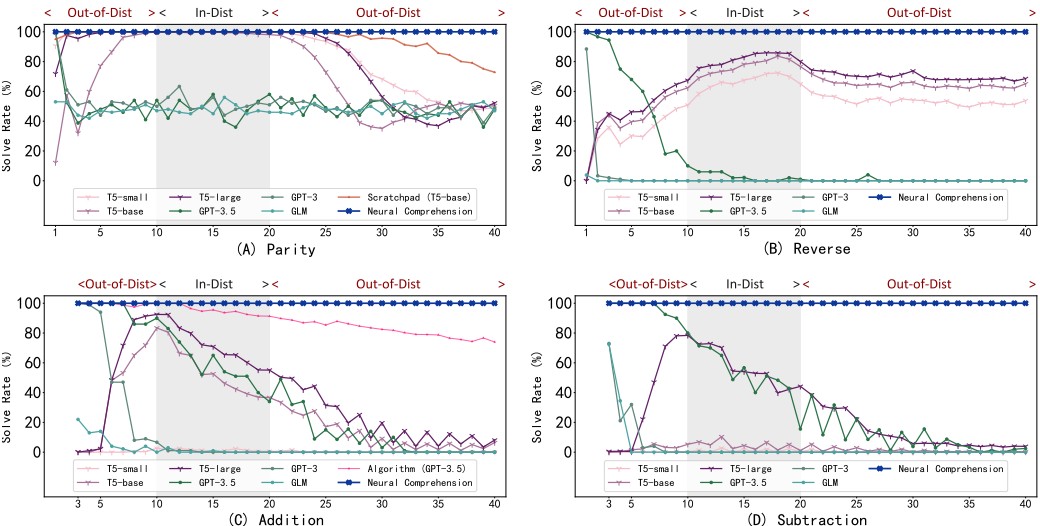

Figure 4: Comparison of Neural Comprehension and other implicit learning-based methods in symbolic operations tasks to test length generalization performance. In this, the T5 model uses the Vanilla Fine-tune method for learning, and LLMs use the Few-shot learning method. In Neural Comprehension, each task has a different CoNN, namely `Parity`, `Reverse`, `Addition`, and `Subtraction`.

| Techniques | In-distribution | Out-of-distribution | Time and Space Complexity | Interpretability |
|---|---|---|---|---|
| Vanilla Fine-tune (For LM) | ✓✓ | ✗ | ✓✓ | ✗ |
| Vanilla Few-shot (For LLM) | ✓ | ✓ | ✓✓ | ✗ |
| Scratchpad (Anil et al., 2022) | ✓✓ | ✓ | ✗ | ✓ |
| Algorithmic (Zhou et al., 2022b) | ✓✓ | ✓ | ✗ | ✓ |
| **Neural Comprehension (Ours)** | ✓✓ | ✓✓ | ✓✓ | ✓✓ |

Table 1: Performance on Symbolic operations tasks of five techniques that language models admit: (1) Vanilla Finetuning, (2) Vanilla Few-shot, (3) Scratchpad (Chain-of-Thought reasoning), (4) Algorithmic (Chain-of-Thought reasoning) and (5) Neural Comprehension. We find that the first four learning-based methods have different modes of failure regarding in and out-of-distribution coverage for symbolic operations. However, Neural Comprehension has strong advantages in terms of length generalization, efficiency, and interpretability. ✗ signifies poor ✓ signifies nontrivial, ✓✓ signifies near-perfect performance. (*) Refers to task-dependency.

We conduct a length generalization experiment (Anil et al., 2022) to examine the distinctions between the Neural Comprehension and learning-based methods, as depicted in Figure 4. Our experimental design encompasses $1000 \times 40$ independent test sets, comprising problems with varying digit lengths from 1 to 40 digits. 10 to 20 digits within the range are provided by us for methods based on implicit learning for training; during the testing phase, this range is called In-Dist. Furthermore, we present results for both Scratchpad (Anil et al., 2022) and Algorithmic (Zhou et al., 2022b) approaches.

The results of our experiment demonstrate that the Vanilla Fine-tune (red lines) method performs optimally on the in-domain (10-20 digit) training set, while its performance deteriorates for both more simplistic and more intricate. This finding suggests that the absence of relevant samples in the training set may cause gradient descent-based language models to underperform on both simpler and more complex tasks. As further discussed in the **appendix D.1**, this phenomenon can be attributed to the inherent generalization limitations of statistical models and the position bias of language models.

Considering the Vanilla Few-shot method (green lines), we determine that its performance is less impacted by the prompt sample range compared to Vanilla Fine-tune. Large language models, which are trained on extensive text corpora, excel at solving more straightforward problems such as symbolic operations within a ten-digit range. Nevertheless, performance remains below par for test sets with more than ten digits, even when prompted with 10-20 digit samples.

Observing CoT-like methods (we use GPT-3.5), including Scratchpad and Algorithmic, unveils their robust length generalization capabilities. Scratchpad works by requiring large language models to record intermediate steps, while Algorithmic employs a similar approach to record the carry operations involved in the addition process. This can be primarily attributed to their proficiency in decomposing complex problems into smaller incremental steps and maintaining intermediate states. However, these methods necessitate substantial computational resources, and extending the length beyond the input limit of the model becomes challenging.

Our study reveals that Neural Comprehension attains remarkably high accuracy in symbolic operations. This implies that Neural Comprehension, unlike conventional methods, does not rely on training data and remains unaffected by discrepancies in input lengths for in-distribution and out-of-distribution data. Consequently, it alleviates the requirement for step-by-step work tracking, and language models with CoNNs only need relatively fewer computational steps to execute sequence operations directly. Encoding rules into neural network modules endows us with greater interpretability, enabling language models to flawlessly perform purely symbolic operation tasks.

## 5.2 SYMBOLIC REASONING

In this section, we investigate the performance of Neural Comprehension in terms of symbolic reasoning capabilities. Our hypothesis is that, although pretrained Language Models (LMs) demonstrate strong language understanding abilities, they lack the capacity to deduce and comprehend rules regarding symbolic reasoning tasks. Thus, we aim to evaluate whether the incorporation of compiled neural networks in the form of CoNNs can address this limitation and improve the LM's symbolic reasoning abilities.

To assess the performance of the rule comprehension component (CoNNs) in symbolic reasoning, we devise an experiment that measures the model's accuracy using intermediate processes and represents them in a "Chain of Thought"-like manner. In doing so, the experiment decomposes language understanding and rule comprehension explicitly into simpler outputs, avoiding the complexities of reasoning and additional error propagation in the models. Ex-

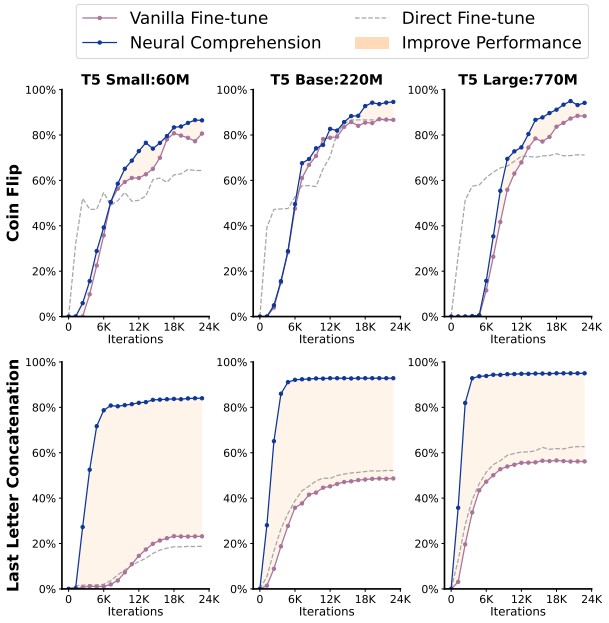

Figure 5: In the iterative process of gradient descent during training. The bleu line represents a language model that incorporates neural comprehension, and the red line represents the original language model. Additionally, we provide Direct, which is a direct prediction of the final result, as a reference.

ample outputs from this approach can be found in **Appendix F**. We observed that neural comprehension improves the symbolic reasoning capabilities of pretrained language models in most cases (Neural Comprehension almost always outperforms Vanilla Fine-tune in Figure 5), and can fit faster. This observation suggests that the introduction of compiled neural networks has a positive impact on pretrained LMs, addressing rule comprehension limitations in symbolic reasoning tasks.

## 5.3 ARITHMETIC REASONING

Arithmetic reasoning serves as a suitable testbed for evaluating language models and their ability to address real-world problems. In this study, we examine the AddSub$^+$ dataset variants that involve different digit lengths, utilizing the `Addition` and `Subtraction` models from the CoNNs family. Notably, the capabilities of Neural Comprehension extend beyond these tasks, as CoNNs can also simulate calculators that support multiplication and division operations, and potentially perform

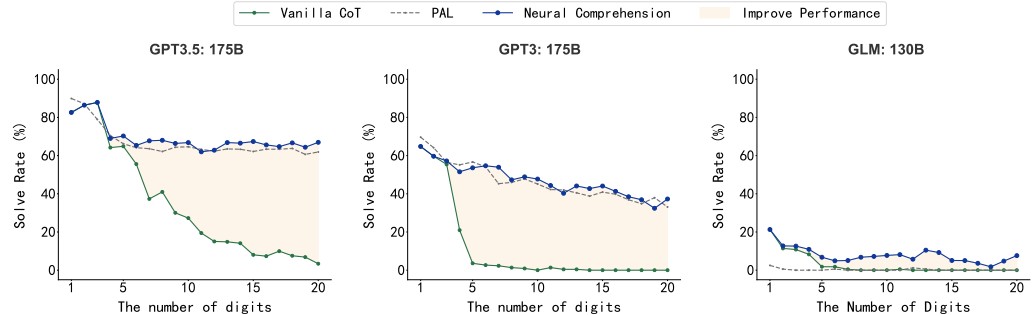

Figure 6: We conducted simulations of the AddSub dataset with varying digits by modifying the "lEquations" parameter. We then tested the performance of three LLMs with and without Neural Comprehension in generating CoT outputs for AddSub⁺. And we reported the solve rates of three LLMs and compared the solve rates of using additional tools (PAL (Gao et al., 2022)).

| Addition | llama-2-7b | llama-2-70b | GLM-130B | Avg |
|---|---|---|---|---|
| CoT | 6.3 | 43.8 | 6.3 | 18.8 |
| PAL | **18.7** | 37.5 | 6.3 | 20.8 |
| NC (Ours) | 12.5 | **43.8** | **7.2** | **21.2** |

| Subtraction | llama-2-7b | llama-2-70b | GLM-130B | Avg |
|---|---|---|---|---|
| CoT | 6.3 | 27.9 | 1.8 | 12.0 |
| PAL | 7.2 | 31.5 | 3.6 | 14.1 |
| NC (Ours) | **9.9** | **32.4** | **4.5** | **15.6** |

| Mixed | llama-2-7b | llama-2-70b | GLM-130B | Avg |
|---|---|---|---|---|
| CoT | 11.1 | 16.7 | 0.0 | 9.3 |
| PAL | 11.1 | 27.8 | 5.6 | 14.8 |
| NC (Ours) | **27.8** | **33.3** | **5.6** | **22.2** |

Table 2: Experiments on the addition and subtraction subset for GSM8K-Hard dataset showing performance comparisons across different models and methods: Only Addition (left), Only Subtraction (center), and Mixed addition and subtraction (right), using Vanilla CoT, PAL, and NC (Neural Comprehension) methods.

linear algebra computations or even in-context learning algorithms that employ backpropagation (Giannou et al., 2023).

To evaluate the impact of Neural Comprehension on arithmetic reasoning, we compare the output of vanilla CoT language models and those incorporating Neural Comprehension, using the vanilla CoT baseline as a reference. As demonstrated in Figure 6 and Table 2, the vanilla CoT model struggles to extrapolate and solve arithmetic problems involving longer digit lengths. However, integrating Neural Comprehension significantly improves the performance of language models on such complex arithmetic tasks. Since we only incorporated the `Addition` and `Subtraction` CoNNs, we attribute the observed performance enhancement to the increased computational accuracy of the language model. For further evidence, we present additional experimental results on widely-used arithmetic reasoning datasets in **Appendix** E.2, which reinforce the benefits of using Neural Comprehension over the vanilla CoT model.

In comparison to language models employing external tools like PAL (Gao et al., 2022), our findings suggest that Neural Comprehension offers greater flexibility for LM. Firstly, by design, it minimizes the necessity for additional processing steps or external tools, leading to an efficient direct computational approach. This contrasts with Tool-based methods that often require additional programming and execution steps, increasing complexity and computational resources. Moreover, CoNN maintains end-to-end differentiability, crucial for models adapting to new data or tasks. In contrast, Tool-based methods are non-differentiable, limiting their adaptability in reinforcement learning settings or tasks with sparse delayed rewards (Chung et al., 2022; Ouyang et al., 2022). Furthermore, CoNN's modularity enhances performance across various model scales, applicable regardless of a language model's ability to call functions, unlike tools only operable in large, additionally code-trained models. Thus, the Neural Comprehension framework's efficiency, unified end-to-end neural network architecture, and extensive applicability constitute its distinct advantages over the Tool-based approach, offering a robust and scalable solution for a multitude of linguistic and computational challenges.

## 5.4 ABLATION AND ANALYSES: MODULE COMBINATION FOR NEURAL COMPREHENSION

Efficiently deploying multiple CoNNs is crucial for achieving exceptional Neural Comprehension performance. As depicted in Figure 7, the amalgamation of distinct CoNNs, tailored for both symbolic and arithmetic reasoning tasks within the language model framework, can lead to remarkable benefits. Similar to ToolFormer (Schick et al., 2023), we combine multiple different CoNNs into

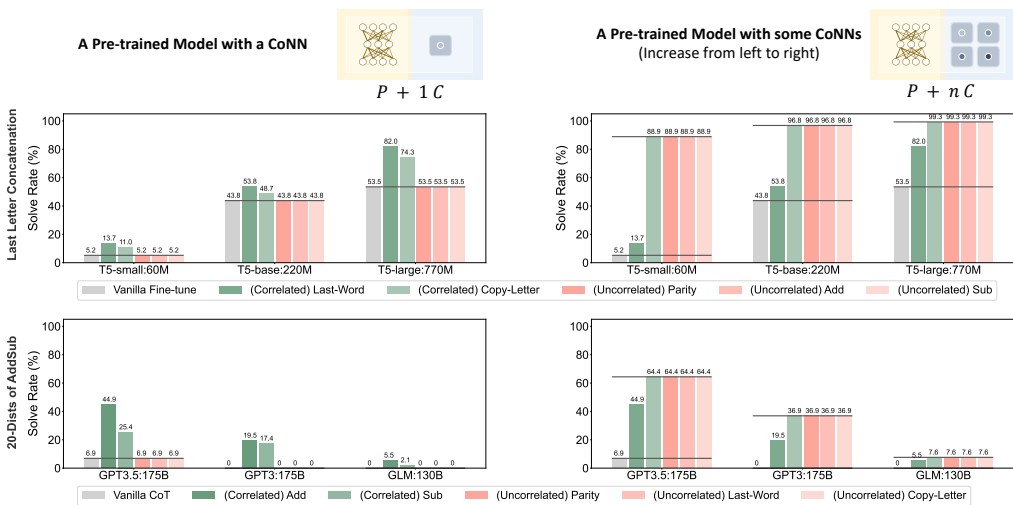

Figure 7: In Neural Comprehension framework, the performance of multiple different module combination is demonstrated. The left side shows the effect of combining a pretrained language model with a CoNN, while the right side shows the impact of combining a language model with multiple CoNNs. For different tasks, we categorize CoNNs as Correlated (green) and Uncorrelated (red), indicating whether the CoNN is related to the current task or not.

one framework, enabling the language model to have multiple capabilities. We conduct experiments on Last Letter Concatenation tass and AddSub$^+$ dataset, which shows the plug-and-play gating mechanism can still well control these CoNNs to output what should be output. It is observed that integrating pertinent CoNNs bolsters the performance of the initial language model, whereas the inclusion of unrelated language models rarely causes detrimental effects, regardless of whether single or multiple CoNNs are combined.

This can be ascribed to the refined design of the Neural Comprehension framework, which ensures the precise execution of assigned tasks by CoNNs without interference from irrelevant modules. Each CoNN module is adept at generating the appropriate output when needed, thereby preventing the emergence of erroneous results from unrelated components. Importantly, as seen in **Appendix B.3**, the parameter count for each CoNN module ranges from 1/1000 to 1/1000000 of that for GPT-3, and the experiments in **Appendix D.3** show that the inference latency in the neural understanding framework only increases by 1%-3% compared to Vanilla.

This observation underscores the remarkable scalability of the Neural Comprehension framework, which possesses the capability to not only accommodate existing knowledge concepts but also assimilate novel ones as the number of CoNNs expands. Theoretically, the integration of tens of thousands of CoNN modules within language models holds the potential to foster a comprehensive understanding of concepts.

## 6 CONCLUSION

We have observed that language models lack an intrinsic comprehension of rule-based concepts and explored how Neural Comprehension can integrate compiled neural networks into the language model framework in a simple and plug-and-play manner. On the one hand, we demonstrated the superiority of our approach over existing learning-based methods, where our method implements comparable improvements to external tools within the neural network framework but does not require additional interpreters. This also enables language models without coding capabilities to possess symbolic manipulation abilities. On the other hand, compared to external tools, gradients can propagate without proxies, allowing better integration and full differentiability. The Neural Comprehension solves the issue of language models themselves being unable to perform robust symbolic operations and providing a foundation for future work on unifying both implicit and explicit learning in language models and facilitating seamless integration.

## REPRODUCIBILITY STATEMENT

All CoNN models mentioned in this paper have been saved in Pytorch format in the **Supplementary Materials**, with dropout set to 0 to ensure deterministic outputs that conform to human-specified rules. The code for the AutoCoNN toolkit and Neural Comprehension framework in this paper can be found in the code portion of the **Supplementary Materials**. Details of all experiments setting referenced in this paper are included in **Appendix** F.1. Detailed descriptions of all tasks, datasets, and baselines mentioned in this paper are provided in **Appendix** F.2. Details of all few-shot prompts referenced in this paper are included in **Appendix** G.

## ACKNOWLEDGEMENTS

This work was supported by the National Key R&D Program of China (No.2022ZD0118501) and the National Natural Science Foundation of China (No.U1936207, No.62376270, No.62171183). Youth Innovation Promotion Association CAS, and OPPO Research Fund.

### ACKNOWLEDGEMENTS FOR COLLEAGUES AND FRIENDS

We appreciate the interest shown in this work by our colleagues and friends:

- Haining Xie, Huanxuan Liao, Jiachun Li, Liang Gui, Pengfan Du, Pengfei Cao, Shaoru Guo, Wangtao Sun, Wenting Li, Xiusheng Huang, Yao Xu, Yifan Wei, Zhao Yang, Zhiqi Wang, Zhongtao Jiang, Zhuoran Jin, Ziyang Huang, who offered enthusiasm and backing.

- Bingmei Sun, Boyuan Jiang, HaoChen Cao, Zixuan Cao, Donghui Li, Dongfang Suze, Ertan Zhuang, Jiajia Li, Mingwei Zhang, Lin Zhang, Mingwen Niu, Min Xiao, Qiaomu Tan, Tianyu Mu, Yuxin Liu, Xiaoyan Yu, Yuke Shi, Yixuan Li, Yang Zhou.

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

## A    DISCUSSION OF LIMITATIONS AND FUTURE WORK

We have presented a novel framework that integrates Compiled Neural Networks (CoNNs) with existing language models to bolster their rule understanding abilities. Although our approach has shown promising performance improvements on symbolic and arithmetic reasoning tasks, there are several limitations and potential avenues for future research that warrant further exploration.

A significant limitation of our current framework lies in the more efficient and natural incorporation of CoNNs into language models. Currently, our method employs a sparse neural network that treats the pretrained language model and CoNNs as separate modules. A more desirable solution is to leverage a dense neural network, simultaneously utilizing the benefits of both components. Examining the large-scale applicability of CoNNs is a beneficial endeavor. Although our experiments have been conducted on a relatively small scale (up to five stacked CoNNs), the advancements and abilities language models may gain from larger-scale combinations of CoNNs remain unclear. Exploring the scalability of our method and the performance advantages of deploying more complex CoNN architectures in language models could provide valuable insights into their potential.

Another promising area of research is the inclusion of explicit knowledge into CoNNs. While the current implementation centers on encoding rules into CoNNs, future work could exploit techniques from knowledge graphs to compile explicit knowledge into language models. This may significantly enhance language models' interpretability and knowledge representation capabilities, potentially resulting in improved performance on an even broader range of tasks.

In conclusion, our work on enhancing language models' rule understanding capabilities through CoNN integration has yielded promising results, albeit with some limitations and remaining challenges. By addressing these areas and extending our approach, we believe that it can ultimately lead to the development of more powerful, interpretable, and knowledge-rich language models.

## B    COMPILED NEURAL NETWORKS

In this section, we will discuss the concept, implementation, and potential of Compiled Neural Networks (CoNN), a type of neural network inspired from previous works on transformers. CoNNs can perform diverse tasks such as computer arithmetic and linear algebra, demonstrating a wide range of applications in LMs and beyond.

### B.1 INTRODUCTION

Transformers have garnered significant attention due to their ability to capture high-order relations and manage long-term dependencies across tokens through attention mechanisms. This enables transformers to model contextual information effectively. Pretrained language models, such as GPT-3 (Brown et al., 2020), exploit contextual learning to invoke various modules for different tasks, like performing arithmetic upon receiving arithmetic prompts. To further enhance rule comprehension in such models, CoNN-based modules are introduced as a part of Neural Comprehension.

Distinct from common models like BERT (Devlin et al., 2018), CoNNs leverage a transformer structure and derive their weights from specialized design rather than pretraining. Each Attention layer and Multilayer Perceptron (MLP) layer in a CoNN represents a specific sequence transformation, leading to a neural network module embodying explicit and interpretable operations.

RASP (Weiss et al., 2021) is a Restricted Access Sequence Processing Language that abstracts the computational model of Transformer-encoder by mapping its essential components, such as attention and feed-forward computation, into simple primitives like select, aggregate, and zipmap. This language enables RASP programs to perform various tasks like creating histograms, sorting, and even logical inference, as demonstrated by Clark et al. (2020).

Tracr (Lindner et al., 2023) serves as a compiler that converts human-readable RASP code into weights for a GPT-like transformer architecture with only a decoder module. The Tracr framework uses JAX to transform RASP-defined code into neural network weights. Our neural reasoning framework employs weights generated by Tracr, which are then converted into PyTorch weights to be compatible with the pretrained language model.

Looped Transformers as Programmable Computers (Giannou et al., 2023) introduces a novel transformer framework that simulates basic computing blocks, such as edit operations on input sequences, non-linear functions, function calls, program counters, and conditional branches. This is achieved by reverse engineering attention and hardcoding unique weights into the model, creating a looped structure. The resulting CoNN can emulate a general-purpose computer with just 13 layers of transformers, and even implement backpropagation-based context learning algorithms, showcasing the approach's vast application prospects.

Overall, the potential applications of CoNNs are extensive, given their capacity to perform a wide array of tasks beyond natural language processing. CoNNs offer increased interpretability and transparency through explicitly defined operations, which is vital in fields such as medical diagnosis and legal decision-making. Additionally, CoNNs can lead to more efficient and effective neural network architectures by reducing pretraining requirements and facilitating improved optimization of network parameters.

### B.2 EXAMPLE

In this subsection, we briefly describe how computational processes can be represented using transformer code and demonstrate how new CoNN weights can be obtained with the aid of the Tracr compiler.

#### B.2.1 PARITY CoNN

In the introduction, we tried to introduce how to perform parity checking on a sequence containing [0 | 1] using a CoNN. Whenever we need to check the sequence, this CoNN can output the completely correct answer.

Figures 8 and 9 present two distinct input sequences, and illustrate the corresponding hidden state and final output obtained after passing through the internal layers of the `Parity` CoNN architecture.

#### B.2.2 REVERSE CoNN

Figures 10 and 11 show the hidden state and output of Reverse CoNN when inputting text. The embedding of CoNN can be customized, so tokens can be either words like 'hello' or individual letters.

THE TRACR CODE OF PARITY CONN

```python
def parity(sop) -> rasp.SOp:
    """Multiply the length of each token."""
    sop = rasp.SequenceMap(lambda x,y: x * y,sop,length).named('map_length')

    """Add each bit."""
    out = rasp.numerical(rasp.Aggregate(rasp.Select(rasp.indices,rasp.indices,rasp.Comparison.
      TRUE).named('Select'),rasp.numerical(rasp.Map(lambda x: x, sop).named('map_length')),
      default=0).named('Aggregate'))

    """Calculate whether the remainder of dividing it by 2 is odd or even."""
    out = rasp.Map(lambda x: 0 if x % 2 == 0 else 1,out).named('Zipmap')

    return out
```

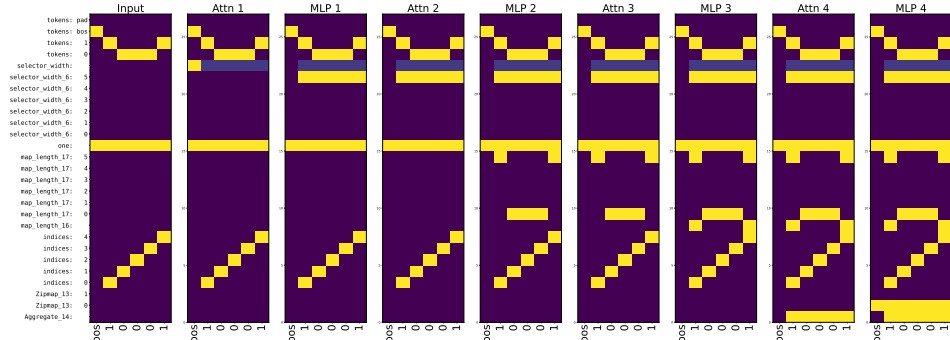

Figure 8: Input the *[1,0,0,0,1]* (target output = 0) for `Parity` CoNN.

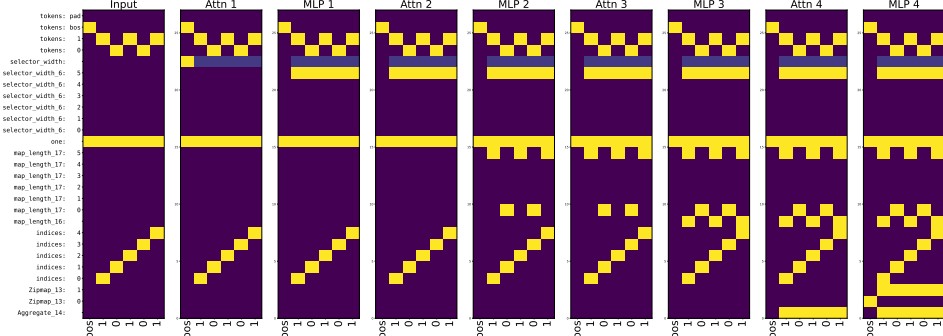

Figure 9: Input the *[1,0,1,0,1]* (target output = 1) for `Parity` CoNN.

### B.2.3    ADDITION CONN

Due to the high complexity of the model, we decided to omit the hidden state transformation for the `Addition` CoNN. However, we have provided code later in the text that will allow for easy implementation of this CoNN. The code includes `add_in_the_same_position` and `add_carry` functions, which are used to calculate the addition and carry of pairs in the CoNN respectively. We divide the entire operation into two models. For the `add_carry` model, we refer to the approach of ALBERT. After the output of the `add_in_the_same_position` model, we cyclically use the `add_carry` model L times, where L is the length of the text, to ensure that all digits can carry. It is important to note that this particular `Addition` CoNN is only capable of performing addition operations on natural numbers.

## THE TRACR CODE OF REVERSE CONN

```python
def reverse(sop) -> rasp.SOp:
    """Get the indices from back to front."""
    opp_idx = (length - rasp.indices).named("opp_idx")

    """opp_idx - 1, so that the first digit of indices = 0."""
    opp_idx = (opp_idx - 1).named("opp_idx-1")

    """Use opp_idx to query indices, get the Select."""
    reverse_selector = rasp.Select(rasp.indices, opp_idx,rasp.Comparison.EQ).named(
        "reverse_selector")

    """Aggregate the reverse_selector and sop"""
    return rasp.Aggregate(reverse_selector, sop).named("reverse")
```

## THE TRACR CODE OF ADDITION CONN

```python
def split(sop, token, index):
    """Match the position of target token"""
    target_position = rasp.Aggregate(rasp.Select(sop, rasp.Map(lambda x: token, sop), rasp.
        Comparison.EQ), rasp.indices)

    """If need to match the front position."""
    if index == 0:
        out = rasp.Aggregate(rasp.Select(rasp.indices, rasp.indices - (length -
            target_position), rasp.Comparison.EQ),
                                sop) # Move the sop on the left side of the token to the far
                                    right.
        return rasp.SequenceMap(lambda x, i: x if i == 2 else "_", out, rasp.categorical(
            rasp.SequenceMap(lambda x, i: 2 if x >= i else 0, rasp.indices, length -
                target_position))) # Use "_" to fill the empty position on the left.

    """If need to match the finally number."""
    else:
        return rasp.SequenceMap(lambda x, i: x if i else "_", sop,
                                rasp.SequenceMap(lambda x, i: 1 if x > i else 0, rasp.indices,
                                    target_position)).named(
            f"shift") # Use "_" to fill the empty position on the left.

def atoi(sop):
    """Converts all text to number, and uses 0 for strings of types other than numbers, It may
        be mixed with 'str' or 'int'.
    """

    return rasp.SequenceMap(lambda x, i: int(x) if x.isdigit() else 0, sop, rasp.indices).
        named(
        "atoi")

def shift(sop):
    """Get the target indices."""
    idx = (rasp.indices - 1).named("idx-1")

    """Use opp_idx to query indices, get the Select."""
    selector = rasp.Select(idx, rasp.indices,
        rasp.Comparison.EQ).named("shift_selector")

    """Aggregates the sops and selectors (converted from indexes)."""
    shift = rasp.Aggregate(selector, sop).named("shift")
    return shift

def add_in_the_same_position(sop):
    x = atoi(split(sop,'+',0)) + atoi(split(sop,'+',1))
    return x

def carry(sop):
    weight = shift(rasp.Map(lambda n:1 if n>9 else 0,sop))

    weight = rasp.Aggregate(rasp.Select(rasp.indices,rasp.indices,lambda key,query:key ==
        query),weight,default=0)
    x = rasp.Map(lambda n:n-10 if n>9 else n,sop)
    return x + weight
```

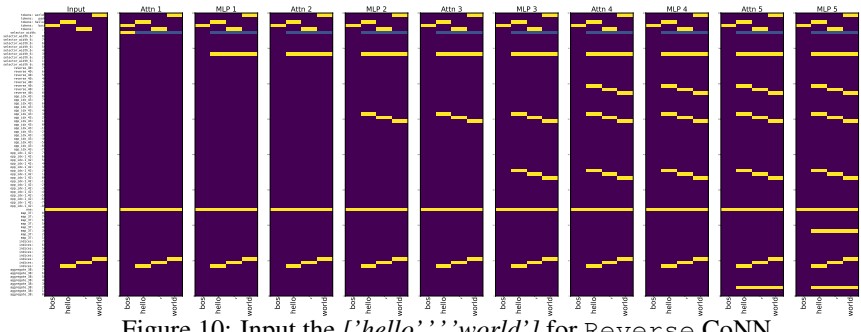

Figure 10: Input the *['hello',',','world']* for `Reverse` CoNN.

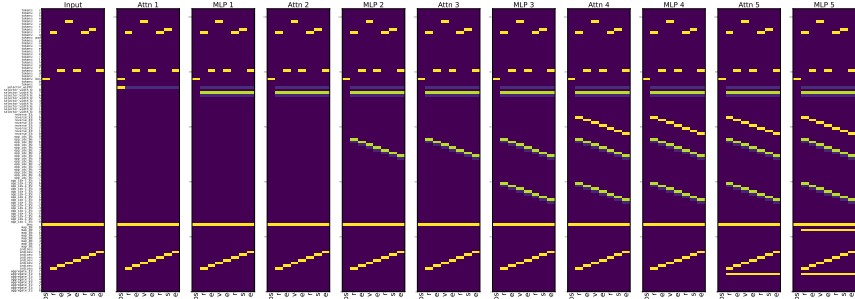

Figure 11: Input the *['r','e','v','e','r','s','e']* for `Reverse` CoNN.

THE TRACR CODE OF SUBTRACTION CONN

```python
def split(sop, token, index):...

def atoi(sop):...

def shift(sop):...

def sub_in_the_same_position(sop):
    x = atoi(split(sop,'-',0)) - atoi(split(sop,'-',1))
    return x

def carry(sop):
    weight = shift(rasp.Map(lambda n:1 if n<0 else 0,sop))
    weight = rasp.Aggregate(rasp.Select(rasp.indices,rasp.indices,lambda key,query:key ==
        query),weight,default=0)
    x = rasp.Map(lambda n:n+10 if n<0 else n,sop)
    return x - weight
```

### B.2.4 SUBTRACTION CONN

The subtraction CoNN is similar to the addition CoNN. First, each digit is subtracted from its corresponding digit, and then it is determined whether to carry over. For ease of experimentation, this subtraction CoNN only supports subtraction of natural numbers where the minuend is greater than the subtrahend.

### B.3 CONN MODEL PARAMETERS

The parameter sizes of all CoNN models used in this work are listed in Table 3. It is noteworthy that even for GPT-3, which has parameters that are orders of magnitude larger, it remains challenging to solve symbolic problems. However, with the use of compiled neural networks, only a small number of parameters are needed to achieve Neural Comprehension.

| Model | Layers | Heads | Vocabulary Size | Window Size | Hidden Size | MLP Hidden Size | # Parameters | Compared to GPT-3 |
|---|---|---|---|---|---|---|---|---|
| `Pariity` | 4 | 1 | 4 | 40 | 132 | 1959 | 2.2M | $\approx 1/100,000$ |
| `Reverse` | 4 | 1 | 28 | 40 | 297 | 1640 | 4.3M | $\approx 1/50,000$ |
| `Last Letter` | 3 | 1 | 28 | 16 | 103 | 32 | 62.6K | $\approx 1/3,000,000$ |
| `Copy` | 1 | 1 | 28 | 16 | 69 | 26 | 8.8K | $\approx 1/20,000,000$ |
| `Add_in_the_same_position` | 7 | 1 | 13 | 40 | 535 | 6422 | 51.8M | $\approx 1/3000$ |
| `Add_Carry` | 3 | 1 | 122 | 40 | 130 | 52 | 117K | $\approx 1/1,500,000$ |
| `Sub_in_the_same_position` | 7 | 1 | 13 | 40 | 535 | 6422 | 51.8M | $\approx 1/3000$ |
| `Sub_Carry` | 3 | 1 | 122 | 40 | 130 | 52 | 117K | $\approx 1/1,500,000$ |

Table 3: We reported on a CoNN with a single function, including its actual parameter size and comparison with the parameters of GPT-3.

### B.4 ENVIRONMENTAL AND HUMAN-CENTRIC BENEFITS OF COMPILED NEURAL NETWORKS

Compiled Neural Networks (CoNNs) address concerns related to the environmental impact of training large models and the need for human-centric computing. CoNN models can reduce energy consumption and carbon emissions by minimizing extensive pretraining and decreasing parameter size, as seen in Table 3. This reduction in computational power and energy requirements makes both the training and inference processes more environmentally friendly. Additionally, Neural Comprehension offers a more interpretable and transparent alternative to conventional deep learning models. CoNN's explicit operation definitions and specialized architecture enable users to comprehend the reasoning behind model decisions, fostering trust and facilitating human-AI collaboration. Increased interpretability also allows for scrutiny of model behavior, promoting the development of fair, accountable, and transparent systems aligned with ethical considerations and human values.'

## C   EXPERIMENT FOR AUTOCONN

### C.1   METHOD

For experts, they may need to spend a lot of time writing code suitable for CoNN, while non-expert users find it hard to obtain or modify CoNN. These issues limit the efficient combination of CoNN and LM, so we utilized the few-shot ability of language models to make the AutoCoNN toolkit (Weng et al., 2023b). In this section, we will show a series of detailed experiments on AutoCoNN to demonstrate this. First, It is the Demo of Auto-CoNN code:

| CoNN Model | Example=1 | Example=2 | Example=5 |
|---|---|---|---|
| `Parity` Model | 5/10 | 10/10 | 10/10 |
| `Reverse` Model | 10/10 | 10/10 | 10/10 |
| `Last Letter` Model | 9/10 | 10/10 | 10/10 |
| `Copy` Model | 10/10 | 10/10 | 10/10 |

Table 4: For each CoNN model, we selected ten groups of models that were judged to be correct by AutoCoNN. We manually evaluated whether these models were indeed correct. The 'Example=x' means that x Examples were provided in the validation stage.

DEMO OF AUTOCONN

```python
from NeuralCom.AutoCoNN import AutoCoNN

INSTRUCT = 'Create_an_SOp_that_is_the_last_letter_of_a_word'
VOCAB = ['a','b','c','d','e','f','g']
EXAMPLE = [[['a','b','c'],['c','c','c']],[['b','d'],['d','d']]]

auto = AutoCoNN()
model,tokenizer = auto(instruct=INSTRUCT,vocab=VOCAB,example=EXAMPLE)
```

Table 5 shows the efficiency comparison between experts and AutoCoNN. This demonstrates that the AutoCoNN toolkit can generate various CoNNs faster. But we also found that for more difficult ones like Addition and Subtraction, it fails to successfully generate, which becomes one of the limitations of AutoCoNN. On the other hand, we tried providing only "Instruct" or "Example" for AutoCoNN to generate[3], and often "Instruct" can generate CoNN with higher accuracy, while "Example" cannot. This shows that giving explicit operational instructions performs better than directly observing data in AutoCoNN.

---

[3]In this experiment, the few-shot samples also contained only one of them

| CoNN Model | Expert's Working Time | Success by AutoCoNN | Can AutoCoNN solve | AutoCoNN (w. Instruct) | AutoCoNN (w. Example) |
|---|---|---|---|---|---|
| `Parity` Model | 1 hours | 8/20 | ✓✓ | 7/20 | 3/20 |
| `Reverse` Model | 0.5 hour | 15/20 | ✓✓ | 16/20 | 11/20 |
| `Last Letter` Model | 0.5 hour | 13/20 | ✓✓ | 12/20 | 10/20 |
| `Copy` Model | 0.2 hour | 17/20 | ✓✓ | 17/20 | 15/20 |
| `Addition` Model | 48 hours | 0/20 | ✗ | 0/20 | 0/20 |
| `Subtraction` Model | 48 hours | 0/20 | ✗ | 0/20 | 0/20 |

Table 5: Comparison between AutoCoNN and Expert Built CoNN. The column 'Expert's Working Time' refers to the time required for a trained engineer to write the CoNN code; 'Success by AutoCoNN' refers to the accuracy of 20 results generated by using GPT-3.5 for diverse decoding; 'Can AutoCoNN solve' refers to whether AutoCoNN can identify suitable CoNN code from the 20 results through validation. It is worth noting that in this experiment, we use sampling decoding with temperature=0.7 to generate 20 different CoNNs codes, which we convert to Pytorch versions of CoNNs models. We report the accuracy of the CoNNs codes through manual (expert) evaluation.

It is difficult for non-expert users to assess the accuracy of the generated code, we automatically utilize the Example information to verify the accuracy of the CoNN model - checking whether the output result of the input sequence is exactly consistent with the Example. The results shown in Table 4 demonstrate that generally 2 Examples are sufficient to select an accurate CoNN model, which means it is very easy for users to use and demonstrate. However, considering the varying difficulty of different tasks, we still suggest non-expert users provide more Examples to ensure the accuracy of the generated CoNN.

## D    EXPERIMENTAL SETTINGS

In this study, we primarily explore the capacity of language models to address symbolic reasoning tasks, concentrating on three areas: symbolic operations, symbolic reasoning, and arithmetic reasoning.

**Symbolic Operations**    Building upon the approaches developed by Anil et al. (2022) and Qian et al. (2022), we examine the following tasks: Parity, Reverse, Addition and Subtraction. These tasks do not require complex text understanding, but only require faithfully implementing symbolic operations and outputting the corresponding results.

**Symbolic Reasoning**    We employ the experimental framework of Wei et al. (2022b) for the two tasks, Last Letter Concatenation and Coin Flip. These tasks require a combination of language understanding and rule comprehension abilities.

**Arithmetic Reasoning**    To evaluate the method's generalization ability from symbolic operations to arithmetic reasoning in addition and subtraction tasks, we use five established arithmetic reasoning datasets: AddSub (Hosseini et al., 2014), SingleEq (Koncel-Kedziorski et al., 2015), MultiArith (Roy & Roth, 2016), GSM8K (Cobbe et al., 2021), and SVAMP (Arkil et al., 2021). Additionally, we introduce the AddSub$^+$ dataset, containing tasks of varying complexity based on the number of digits involved in arithmetic operations, ranging from 1-digit addition to 20-digit addition/subtraction tasks.

## E    SUPPLEMENTARY EXPERIMENT

### E.1    THE EFFECT OF TRAINING DATA SCALE ON LENGTH GENERALIZATION OF GRADIENT-BASED MODELS

To investigate the impact of training data scale on out-of-distribution (OOD) performance, we conducted experiments using the T5-large model with varying amounts of in-distribution training data. The experimental setup closely followed that of **Main Figure 3**, utilizing numbers with 10 to 20 digits as the training set but varying the number of training examples between 1 million and 15 million. The peak validation set performance for each experiment is reported in Figure 12.

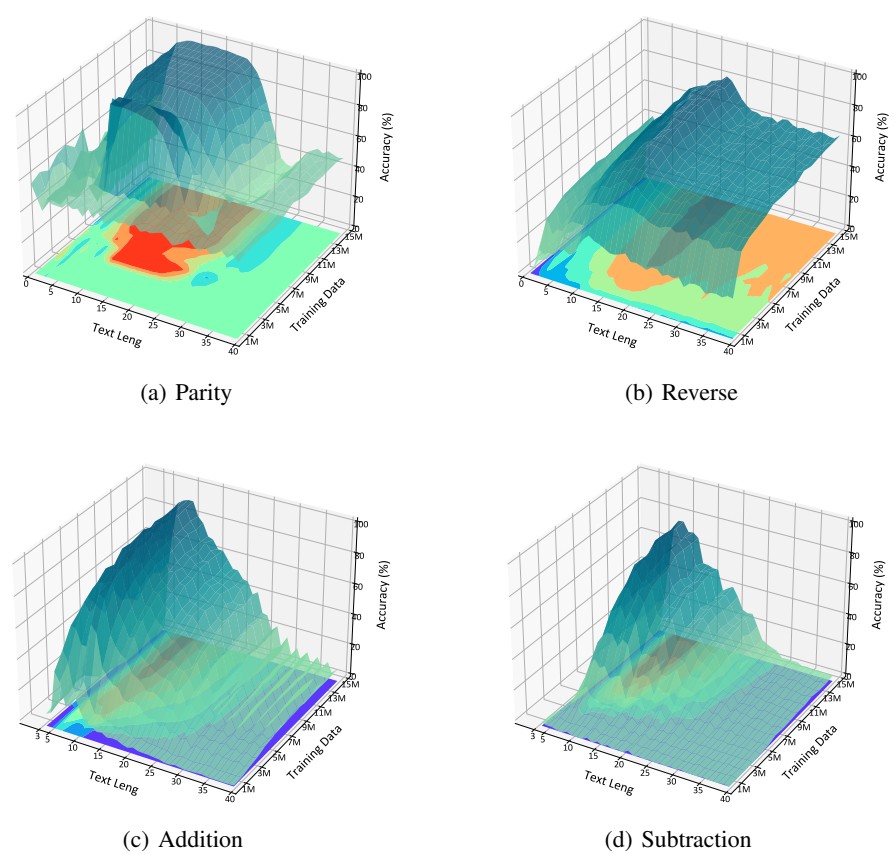

(a) Parity

(b) Reverse

(c) Addition

(d) Subtraction

Figure 12: Length Generalization Performance of Language Models with Different Dataset Sizes.

The results in Figure 12 show that increasing the scale of the In-Dist training data leads to only marginal improvements in OOD performance. This finding is discouraging, suggesting that gradient-based language models face challenges in capturing the true underlying meaning of symbols and their transformation rules based on the data distribution alone.

## E.2 REAL-WORLD ARITHMETIC REASONING TASKS

| | Method | GSM8K | SingleEq | AddSub | MultiArith | SVAMP | **Average** |
|---|---|---|---|---|---|---|---|
| | Previous SOTA (Fintune) | $35^a/57^b$ | $32.5^c$ | $94.9^d$ | $60.5^e$ | $57.4^f$ | - |
| | GPT-3 Standard | 19.7 | 86.8 | 90.9 | 44.0 | 69.9 | 62.26 |
| GPT-3 (175B) `code-davinci-001` | CoT | 13.84 | 62.02 | 57.22 | 45.85 | 38.42 | 43.47 |
| | CoT + Neural Comprehension | $13.95_{(+0.11)}$ | $62.83_{(+0.81)}$ | $60.25_{(+3.03)}$ | $45.85_{(+0.0)}$ | $38.62_{(+0.2)}$ | $44.30_{(+0.83)}$ |
| GPT-3.5 (175B) `code-davinci-002` | CoT | 60.20 | 91.01 | 82.78 | 96.13 | 75.87 | 81.20 |
| | CoT + Neural Comprehension | $60.42_{(+0.22)}$ | $91.01_{(+0.0)}$ | $82.78_{(+0.0)}$ | $96.13_{(+0.0)}$ | $76.09_{(+0.22)}$ | $81.29_{(+0.09)}$ |

Table 6: Problem solve rate (%) on arithmetic reasoning datasets. The previous SoTA baselines are obtained from: (a) GPT-3 175B finetuned (Cobbe et al., 2021); (b) GPT-3 175B finetuned plus an additional 175B verifier(Cobbe et al., 2021); (c) Hu et al. (2019); (d) Roy & Roth (2016); (e) Roy & Roth (2016); (f) Amini et al. (2019); (f) Pi et al. (2022)

As model parameters, training calculations, and dataset sizes have increased, language models have gained new capabilities (Srivastava et al., 2022; Wei et al., 2022a; Weng et al., 2024), such as Machine Translation (Zhao et al., 2023; Li et al., 2024), complex QA (Zhu et al., 2022; Daull et al., 2023; Zhu et al., 2023), Multimodal QA (Wang et al., 2023a; Li et al., 2023; Weng & Li, 2023), coding (Li et al., 2022; Nijkamp et al., 2022), few-shot learning (Brown et al., 2020; Perez et al., 2021), medical diagnosis (Li et al., 2021a; Xia et al., 2022; Weng et al., 2023a), and chain of thought (Wei et al., 2022b; Weng et al., 2023c).

In Table 6, we compared Vanilla CoT with the Neural Comprehension framework for arithmetic reasoning tasks. We integrated the `Addition` and `Subtraction` CoNNs with LLMs and observed improved performance across several tasks. This suggests that the proposed Neural Comprehension framework can compensate for the difficulties faced by large-scale language models in computational tasks. Nevertheless, the performance improvement is not as significant due to the choice of specific CoNN models to ensure clarity in our experiments. Designing CoNN models to support more general arithmetic tasks could potentially yield more substantial improvements. In addition, since the Neural Comprehension framework improves the gap between the data distribution learned by the language model during training through gradient descent and the real rules, it can also be combined with some existing logical improvements to language models, including self-consistency (Wang et al., 2023b), least-to-most (Zhou et al., 2022a), self-improve (Huang et al., 2022), and self-verification (Weng et al., 2023c). It can also be combined with some zero-shot methods (Kojima et al., 2022; Zhang et al., 2023).

To further evaluate the effectiveness of the Neural Comprehension framework, Table 7 presents the results of fine-tuning T5 models with `Addition` and `Subtraction` CoNN on the GSM8K training dataset. The comparison of three different-sized models reveals that the framework can model deterministic rules defined by humans, thus avoiding the uncertainty associated with gradient descent learning from data distribution.

| Method | T5-small | T5-base | T5-large |
|---|---|---|---|
| Origin | 1.74 | 1.52 | 3.87 |
| Neural Comprehension | 1.82 | 1.59 | 4.02 |
| Ours Improve | +0.08 | +0.07 | +0.15 |

Table 7: The test set problem-solving rate (%) of the T5 model on the GSM8K dataset.

### E.3 THE EFFICIENCY OF NEURAL COMPREHENSION

| Model | Params | Task | Vanilla | | Neural Comprehension | | $\delta_{\text{Time}}$ | |
|---|---|---|---|---|---|---|---|---|
| | | | GPU | CPU | GPU | CPU | GPU | CPU |
| T5-small | 60M | Coin Flip | 5.280s | 5.720s | 5.431s | 5.872s | 0.151s (2.86%) | 0.152s (2.66%) |
| T5-base | 220M | Coin Flip | 7.865s | 13.767s | 8.010s | 13.939s | 0.145s (1.84%) | 0.172s (1.25%) |
| T5-large | 770M | Coin Flip | 14.055s | 32.953s | 14.194s | 33.120s | 0.139s (0.99%) | 0.167s (0.51%) |
| T5-small | 60M | Last Letter Concatenation | 16.233s | 28.309s | 16.744s | 28.720s | 0.511s (3.15%) | 0.411s (1.45%) |
| T5-base | 220M | Last Letter Concatenation | 28.912s | 55.660s | 29.426s | 56.087s | 0.514s (1.78%) | 0.427s (0.77%) |
| T5-large | 770M | Last Letter Concatenation | 49.584s | 103.739s | 50.066s | 104.134s | 0.482s (0.97%) | 0.395s (0.38%) |

Table 8: In Neural Comprehension framework, the inference latency comparison of the T5 model.

To evaluate the efficacy of Neural Comprehension, we conducted further experiments comparing the inference latency of both the Vanilla and Neural Comprehension frameworks on an equal number of sequences and equal sequence lengths using GPU and CPU configurations. We employed a batch size of 1 and assessed the inference latency of Neural Comprehension in conjunction with various T5 model sizes across two symbolic inference tasks to ascertain efficiency. The full results are detailed in Table 8.

Our findings reveal that implementing Neural Comprehension increases computational requirements, primarily attributed to the supplementary parameter volume and computational demands introduced by CoNNs. However, as the scale of pretrained language models expands, the proportion of $\delta_{\text{Time}}$ within the Neural Comprehension framework progressively diminishes, particularly for larger language models.

# F Implementation and Details

| Model | Model Creator | Modality | Version | # Parameters | Tokenizer | Window Size | Access |
|---|---|---|---|---|---|---|---|
| T5-small | Google | Text | T5.1.0 | 60M | T5 | 512 | Open |
| T5-base | Google | Text | T5.1.0 | 220M | T5 | 512 | Open |
| T5-large | Google | Text | T5.1.0 | 770M | T5 | 512 | Open |
| GLM-130B | Tsinghua University | Text | GLM-130B | 130B | ICE | 2048 | open |
| GPT-3 | OpenAI | Text,Code | code-davinci-001 | 175B* | GPT-2 | 2048 | limited |
| GP-3.5 | OpenAI | Text,Code,Instruct | code-davinci-002 | 175B* | GPT-2 | 8096 | limited |
| GPT-4 | OpenAI | Text,Code,Instruct,... | gpt-4 | 1750B* | GPT-2 | 8000 | limited |

Table 9: Models. Description of the models evaluated in this effort: provenance for this information is provided in models. ∗ indicates that we believe the associated OpenAI models are this size, but this has not been explicitly confirmed to our knowledge.

In this section, we provide a detailed description of the experimental setup from a model and dataset perspective, ensuring repeatability of our experiments.

## F.1 Model

Our experiments primarily involve the T5, GPT, and GLM-130B families of models. Neural Comprehension framework supports seamless integration with language models having decoder structures, regardless of the scale of the language model. We fine-tune the T5 models, while the larger models with over 10 billion parameters are used for few-shot In-context learning. Table 9 presents a comprehensive list of all models used in our experiments[4].

### F.1.1 Fine-tuning

For the T5 models, we employ the standard fine-tuning approach using the pretrained models as a starting point. We follow the pre-processing steps in the T5 original paper, which involves set the input text max length to 150 and using the tokenizer to process the data. We use a batch size of 64 for all models and the Adafactor optimizer (Shazeer & Stern, 2018) with a learning rate of $1 \times 10^{-4}$. The models are trained for a maximum of 20 epochs. We use a cosine learning rate schedule with a warm-up phase comprising 5% of the total number of training steps. We employ a dropout rate of 0.1 during training to mitigate overfitting. Our experiments utilize the PyTorch framework (Paszke et al., 2019) for training and inference. Table 5 displays the parameter settings for the T5 models during training, which is conducted on four NVIDIA A6000 GPUs with 48GB of memory each.

| Training Setting | Configuration |
|---|---|
| optimizer | Adafactor |
| base learning rate | $1 \times 10^{-4}$ |
| weight decay | $2 \times 10^{-5}$ |
| decay rate | -0.8 |
| optimizer eps | $(1 \times 10^{-30}, 2 \times 10^{-3})$ |
| batch size | 64 |
| training epochs | 20 |
| gradient clip | 1.0 |

Table 10: Training Setting

In Table 10, we list the hyperparameters used to train the T5 model. We carefully selected these parameters to ensure that the within-distribution validation accuracy roughly converged. We report all peak validation set results, and in every experiment we ran, we found that within-distribution validation accuracy monotonically increased during training iterations (until it approached 100%), and we never observed overfitting. This may be due to the regular nature of the tasks we considered in the paper. We followed the Anil et al. (2022)'s setup and did not use OOD performance in model selection, as this would constitute "peaking at the test conditions". Regarding the number of training iterations, we also tried training the T5 model with more iterations in the addition task, but this did not lead to substantial differences in OOD performance (i.e., it was still equally poor).

### F.1.2 Few-shot In-Context Learning

For the few-shot context learning on GPT and GLM-130B models, we employ an approach inspired by the recent work on GPT-3 (Brown et al., 2020). In this methodology, the models are provided a

---

[4]The specific configurations of the GPT series of models can be found at `https://platform.openai.com/docs/model-index-for-researchers/model-index-for-researchers`

PARITY DATASET

```python
def generate_parity_data(n):
    data = []
    for _ in range(100000):
        input_str = ''.join(str(random.randint(0, 1)) for _ in range(n))
        label = sum(int(x) for x in input_str) % 2
        data.append({'input': input_str, 'label': label})
    return data

parity_data = {}
for n in range(1, 41):
    parity_data[n] = generate_parity_data(n)
```

context consisting of a few examples of the input-output pairs in natural language format. Following this, the models are expected to generalize and perform well on the task without any explicit fine-tuning. We carefully design the context to include diverse examples that represent the range of input types and required model reasoning. Importantly, we limit the context length to be within the maximum token limits of the models. For instance, GPT-3 has a token limit of 2048. Due to limited access to the GPT family of models, we utilize the official API for these experiments [5]. .For the GLM-130B, we employ the FasterTransformer framework to set up local inference with INT4 on eight NVIDIA GeForce RTX 3090 GPUs with 24GB of memory each.

To compare with CoT and PAL which experimented on GPT-3 series models, we simulated Neural Comprehension (NC) within the constraints of the API access. We treated the API's output as if it were part of the Neural Comprehension structure. This involved a simulated gating mechanism, where the output from the API was corrected using CoNNs form left to right, and then the adjusted response (Truncate the text after the altered text.) was changed into the API' input for continued generation. This simulation was to ensure that the benefits of NC could be compared fairly with the existing results from PAL.

### F.2  TASKS AND DATASET

In this paper, all data sets related to length generalization consist of independent data sets with the same number of digits but different lengths, and each digit in the test set is unique. Therefore, there may be slight fluctuations between data sets of different lengths, but the overall trend is generally clear. To further illustrate the differences between data sets of different lengths, the following examples are provided:

$$\textbf{Parity:} \quad \underbrace{110}_{\text{Length} = 3} = 0 \qquad \underbrace{101100}_{\text{Length} = 6} = 1 \qquad \underbrace{010001110101}_{\text{Length} = 12} = 0$$

$$\textbf{Reverse:} \quad \underbrace{abc}_{\text{Length} = 3} = cba \qquad \underbrace{figure}_{\text{Length} = 6} = erugif \qquad \underbrace{accomplished}_{\text{Length} = 12} = dehsilpmocca$$

$$\textbf{Addition:} \quad \underbrace{1 + 2}_{\text{Length} = 3} = 3 \qquad \underbrace{18 + 245}_{\text{Length} = 6} = 263 \qquad \underbrace{48864 + 964315}_{\text{Length} = 12} = 1013179$$

$$\textbf{Arithmetic Reasoning:} \quad \text{Joan found } \underbrace{6546634574688499}_{\text{Length} = 16} \text{ seashells and Jessica found}$$

$$\underbrace{3855196602063621}_{\text{Length} = 16} \text{ seashells on the beach. How many seashells did they find}$$

together ?

#### F.2.1  DATA GENERATION DETAILS

**Synthetic Parity Dataset:** We sample instances of lengths 1-40 from a uniform Bernoulli distribution. We first uniformly sample the number of ones, and then randomly shuffle the positions of each one within a fixed-length bitstring. For the experiments in **Figure** 5.1, we train T5 on lengths 10-20, with 99000 training samples per bit. For all methods, we test each bit using 1000 samples.

---

[5]OpenAI's API:https://openai.com/api/

REVERSE DATASET

```python
reverse_data = {}
for n in range(1, 41):
    reverse_data[n] = []
    for _ in range(100000):
        word = ''.join(random.choice(string.ascii_lowercase) for _ in range(n))
        reverse_data[n].append({'input':word,'label':''.join(list(reversed(word)))})
```

ADDITION DATASET

```python
# Generate two random n-digit numbers and their sum
def generate_additive_example(n):
    data = []
    k = (n - 1) % 2
    n = (n - 1) // 2 if n > 2 else n

    for _ in range(100000):
        a = random.randint(10**(n+k-1), 10**(n+k) - 1)
        b = random.randint(10**(n-1), 10**n - 1)
        data.append({'input': str(a)+'+'+str(b), 'label': a + b})
    return data

# Generate an additive data set for digits ranging from 3 to 40
additive_data = {}
for n in range(3, 41):
    additive_data[str(n)] = generate_additive_example(n)
```

**Synthetic Reverse Dataset:** We selected a dataset of instances with lengths ranging from 1 to 40. For the experiments in **Figure** 5.1, the training set consists of 99000 samples each for lengths 10-20. Each input is a randomly generated word string of specified length, where the letters are selected uniformly at random from a set of 26 letters using a Bernoulli distribution (without necessarily having any actual meaning), and all letters are converted to lowercase. The test set for lengths 1-40 contains at least 1000 test samples.

**Synthetic Addition and subtraction Dataset:** Addition and subtraction are fundamental arithmetic operations that are commonly taught in primary school. To generate the dataset, we takes as input the number of digits $n$ and returns a list of 100000 examples. The function first calculates the remainder $k$ when $(n-1)$ is divided by 2, and then divides $(n-1)$ by 2 if n is greater than 2, else it sets n to itself. This ensures that the length of the first number is either equal to or one digit longer than the length of the second number. The function then generates 100000 examples using randomly generated numbers. Specifically, it generates two numbers a and b where a is a random integer between $10^{(n+k-1)}$ and $10^{(n+k)} - 1$, and b is a random integer between $10^{(n-1)}$ and $10^n - 1$. It then appends each example to the list data in the form of a dictionary with the input as the string "a+b" and the label as the sum a+b.

For the experiments in Figure **Figure** 1, we provided 99000 training data examples for addition with numbers ranging from 3 to 10 digits in length for the T5 model. For the GPT-3.5 and GPT-4 models, we provided 8 few-shot samples within 10 digits. We evaluated the performance of all three models on numbers ranging from 3 to 30 digits in length, with 1000 test samples per digit. On the other hand, for the experiments in Figure **Figure** 5.1, we provided 99000 training data examples for addition with numbers ranging from 10 to 20 digits in length for the T5 model. For the GPT-3.5 and GPT-4 models, we provided 8 few-shot samples within the range of 10 to 20 digits. We evaluated the performance of all three models on numbers ranging from 3 to 30 digits in length, with 1000 test samples per digit.

For subtraction, we use a similar approach.

### F.2.2 SYMBOLIC REASONING DATASET

**Coin Flip:** We followed Wei et al. (2022b)'s setup and randomly concatenated first and last names from the top 1000 names in the census data (https://namecensus.com/) to create the <NAME> token. In our work, flipping a coin corresponds to 1 and not flipping a coin corresponds to 0. To make the inputs as close to English as possible without using too many symbols, we used the sentence models "<NAME> flips the coin." and "<NAME> does not flip the coin." to represent

SUBTRACTION DATASET

```python
# Generate two random n-digit numbers and their minus
def generate_minus_example(n):
    data = []
    k = (n - 1) % 2
    n = (n - 1) // 2 if n > 2 else n
    for _ in range(100000):
        a = random.randint(10**(n+k-1), 10**(n+k) - 1)
        b = random.randint(10**(n-1), 10**n - 1)
        if a > b:
            data.append({'input': str(a)+'-'+str(b), 'label': a - b})
        else:
            data.append({'input': str(b)+'-'+str(a), 'label': b - a})
    return data

# Generate an subtraction data set for digits ranging from 3 to 40
minus_data = {}
for n in range(3, 41):
    minus_data[str(n)] = generate_minus_example(n)
```

COIN FILP DATASET

```python
dataset = []
for i in range(500):
    # randomly choose two names from the name_list
    for o in range(2,5):
        sentence = 'A coin is heads up.'
        label = []
        for time in range(o):
            name = random.sample(names, k=1)[0]

            # randomly choose whether to flip the coin or not
            flip = random.choice([True, False])

            # generate the statement and label based on whether the coin was flipped or not
            if flip:
                sentence += f" {name.capitalize()} flips the coin."
                label.append(1)
            else:
                sentence += f" {name.capitalize()} does not flip the coin."
                label.append(0)
        sentence += ' Is the coin still heads up?'

        dataset.append({'question':sentence, 'answer':{0:'yes',1:'no'}[sum(label)%2]})
```

whether the coin was flipped or not. This task is similar to the parity task, but requires further semantic understanding. We constructed a training set of 1500 samples, with 500 samples for each of 2-4 coin flips. For the test set, we selected 100 non-overlapping samples for each of 2-4 coin flips, and evaluated the model every 5 steps.

**Last Letter Concatenation:** We followed Wei et al. (2022b)'s setup and randomly concatenated first and last names from the top 1000 names in the census data to create the <NAME> token. This task requires the model to connect the last letter of each word in a concatenated name. This task requires Neural Comprehension of rules in two aspects. First, it requires the model to correctly identify the last letter of each word. Second, it requires the model to concatenate all the last letters of the words together. We concatenated 2-5 first or last names, and constructed a training set of 1500 samples, with 500 samples for each name length of 2-4. For the test set, we selected 100 non-overlapping samples for each name length of 2-4, and evaluated the model every 5 steps.

### F.2.3 ARITHMETICAL REASONING DATASET

In Table 11, we summarize the information of all arithmetic reasoning datasets used in this work. We provide the links to access these datasets:

- **GSM8K**: https://github.com/openai/grade-school-math
- **SingleEq**: https://gitlab.cs.washington.edu/ALGES/TACL2015
- **AddSub**: https://www.cs.washington.edu/nlp/arithmetic
- **MultiArith**: http://cogcomp.cs.illinois.edu/page/resource_view/98

| Dataset | Number of samples | Average words | Answer Format | Lience |
|---|---|---|---|---|
| GSM8K | 1319 | 46.9 | Number | MIT License |
| SingleEq | 508 | 27.4 | Number | MIT License |
| AddSub | 395 | 31.5 | Number | Unspecified |
| MultiArith | 600 | 31.8 | Number | Unspecified |
| SVAMP | 1000 | 31.8 | Number | MIT License |

Table 11: Arithmetical Reasoning Dataset Description.

- **SVAMP**: https://github.com/arkilpatel/SVAMP

# G  SOME EXAMPLES OF NEURAL COMPREHENSION

In this section, we will use gray font to represent the task input, yellow font to represent the neural network output during training, and blue background to represent the neural network output during generated.

## G.1  SYNTHETIC SYMBOLIC

Q: 1011001010 A: 1
Q: 01111011000 A: 0
Q: 1010011001110 A: 1
Q: 10000001001001 A: 0
Q: 110100011110001 A: 0
Q: 1110011001010110 A: 1
Q: 1100000111011000101 A: 1
Q: 01100000110110010001 A: 0
———(LLM's few-shot prompt)———
Q: 011110001010101101011 A:  0

Table 12:  The example of Parity

Q: neofascism A: msicsafoen
Q: betaquinine A: eniniuqateb
Q: corediastasis A: sisatsaideroc
Q: ferroelectronic A: cinortceleorref
Q: cryoprecipitation A: noitatipicerpoyrc
Q: cryofibrinogenemia A: aimenegonirbifoyrc
Q: chemocarcinogenesis A: sisenegonicracomehc
Q: ponjpcdqjuuhiviojmby A: ybmjoivihuujqdcpjnop
————-(LLM's few-shot prompt)————-
Q: helloworldhellochina A:  anihcollehdlrowolleh

Table 13:  The example of Reverse

Q: 82637+3058 A: 85695

Q: 58020+96632 A: 154652

Q: 717471+58704 A: 776175

Q: 298309+702858 A: 1001167

Q: 1061462+2623780 A: 3685242

Q: 58720970+61609034 A: 120330004

Q: 364920479+78861480 A: 443781959

Q: 6050330002+211065324 A: 6261395326

————-(LLM's few-shot prompt)————-

Q: 20021012+20021004 A:  40042016

Table 14: The example of Addition

Q: 82637-3058 A: 79579

Q: 96632-58020 A: 38612

Q: 717471-58704 A: 658767

Q: 702858-298309 A: 404549

Q: 2623780-1061462 A: 1562318

Q: 68720970-61609034 A: 7111936

Q: 364920479-78861480 A: 286058999

Q: 6050330002-211065324 A: 393967676

————-(LLM's few-shot prompt)————-

Q: 20021012-20021004 A:  8

Table 15: The example of Subtraction

## G.2 SYMBOLIC REASONING

A coin is heads up. Devin flips the coin. Maxwell does not flip the coin.

James flips the coin. Kenneth flips the coin. Is the coin still heads up?

1 0 1 1 ->  1

A coin is heads up. Ira flips the coin. Danny does not flip the coin.

Horace flips the coin. Is the coin still heads up?

1 0 1 ->  0

Table 16: The example of Coin Flip

| |
|---|
| Take the last letters of the words in Elias Earnest Milton and concatenate them. |
| The last letter of Elias -> s  The last letter of Earnest -> t |
| The last letter of Milton -> n  The answer is  stn |
| Take the last letters of the words in Randolph Weldon Olin Robbieänd concatenate them. |
| The last letter of Randolph -> h  The last letter of Weldon -> n |
| The last letter of Olin -> n  The last letter of Robbie -> e  The answer is  hnne |

Table 17: The example of Last Letter Concatenation

## G.3 ARITHMETICAL RESONING

| |
|---|
| Joan found 65466345746884996 seashells and Jessica found 38551966020636213 |
| seashells on the beach . How many seashells did they find together ? |
| Joan started with 65466345746884996 seashells. She gave some to Sam. So: |
| 65466345746884996 - 38551966020636213 =  2691437972624878 |
| The answer is 2691437972624878 |

Table 18: The example of Arithmetirc Reasoning

