# OpenReview forum: "Mastering Symbolic Operations: Augmenting Language Models with Compiled Neural Networks"
_ICLR.cc/2024/Conference — ICLR 2024 poster_

### Official Review · Reviewer_uMuv · 2023-10-29

**Soundness:** 3 good
**Presentation:** 2 fair
**Contribution:** 3 good
**Rating:** 6
**Confidence:** 4

**Summary:**

This paper proposes a method called "Neural Comprehension" to augment language models with the ability to perform symbolic reasoning and operations more robustly. The key idea is to incorporate compiled neural networks (CoNNs) that encode rules explicitly into the language model architecture.

The authors argue that standard language models struggle with symbolic tasks like arithmetic due to relying on statistical patterns learned implicitly from data. In contrast, CoNNs directly compile rules like addition into neural network modules using specialized attention weights. By integrating CoNNs in a plug-and-play manner, language models can leverage both their natural language understanding and the symbolic reasoning of CoNNs.

The proposed Neural Comprehension framework uses a gating mechanism to determine when to invoke a CoNN module versus the original language model decoder. Experiments on symbolic operations and arithmetic tasks demonstrate substantially improved performance over baseline methods.

**Strengths:**

(1) Proposes a novel way to impart symbolic reasoning abilities in language models using CoNNs. Addresses a clear limitation of standard LMs.

(2) Achieves strong performance improvements on multiple symbolic and arithmetic datasets over baselines. Shows the benefit of combining statistical and symbolic knowledge.

(3) CoNN integration does not require tool APIs. Comparable to tool-based methods while being end-to-end differentiable and not needing code generation.

(4) Reasoning process is more interpretable compared to black-box LM predictions. Outputs show step-by-step reasoning.

**Weaknesses:**

(1) CoNN design and integration could be non-trivial for more complex reasoning tasks, e.g. compositional arithmetic reasoning tasks. Gating mechanism may work fine if we just want to add one skill. However, it will be hard to scale to thousands of skills for a specific language model and this exploration remains limited. Developing a specific model for a specific task involves significant human efforts and we want to build a model that can have multiple skills.

(2) Although this paper claims it does not require tool API, in my opinion, this is not free and has very large costs. If a calculator is there, I think we should use it instead of building neural networks mimicking these tools. API is also required for internet-based retrieval and data analysis (OpenAI Code Interpreter ), etc. I think adding a python api is simpler than adding a neural module.

(3) Experimental results shown in Table 5 on real-world reasoning tasks, e.g. GSM8K, are similar to baselines without significant improvements.

**Questions:**

See Weaknesses.

---

> ### Author Response · Authors · 2023-11-15
> **Rebuttal by Authors: 1**
>
> **We greatly appreciate the reviewers for their constructive feedback and insightful comments on our work. We will address the identified weaknesses and concerns to clarify the contributions and impact of our work.**
>
> > (1) CoNN design and integration could be non-trivial for more complex reasoning tasks, e.g. compositional arithmetic reasoning tasks. Gating mechanism may work fine if we just want to add one skill. However, it will be hard to scale to thousands of skills for a specific language model and this exploration remains limited. Developing a specific model for a specific task involves significant human efforts and we want to build a model that can have multiple skills.
>
> We recognize the importance associated with integrating CoNNs for a vast array of tasks. In the right part of Figure 7 in the main text, we try combining five different CoNNs on the symbolic reasoning task LLC and arithmetic reasoning task AddSub, where some are relevant (e.g. the Addition Model and Subtraction Model needed for the AddSub task) and others irrelevant (e.g. the Parity Model, Copy-Letter Model, and Last-Word Model for the AddSub task).
>
> **Table D-1:** Result of Figrue 7 for 20-Dists of AddSub+
> | **GSM8K-Hard (Subtraction)** | Vanilla CoT | Add | Sub | (Uncorrelated) Parity | (Uncorrelated) Laster-Word |  (Uncorrelated) Copy-Letter |
> |-|-|-|-|-|-|-|
> |GPT-3.5| 6.9 | 44.9 | 64.4 | 64.4 | 64.4 | 64.4 |
> |-|-|-|-|-|-|-|
> |GPT-3| 0 | 19.5 | 36.9 | 36.9 | 36.9 | 36.9 |
> |-|-|-|-|-|-|-|
> |GLM-130B| 0 | 5.5 | 7.6 | 7.6 | 7.6 | 7.6 | 7.6 |
>
> In Table D-1, **from left to right**, we gradually add additional CoNNs, where Add and Sub are Correlated, and the rest are Uncorrelated. For GPT-3.5, GPT-3 and GLM-130B, adding Add and Sub leads to better performance than without adding them, while adding Uncorrelated CoNNs does not degrade performance. **Our experiments have proven that the simple plug-and-play gating mechanism is sufficient to manipulate multiple different groups of CoNNs with almost no interference from irrelevant models. In fact, the Toolformer[1] that needs additional training likewise has five API invocation skills:**
>
> Then, to generalize to new works, our proposed AutoCoNN toolkit facilitates the automated creation of CoNNs with minimal human intervention. It utilizes Large Language Models' intrinsic code-writing and deductive abilities to generate task-specific CoNNs based on given "Instruct" and a few "Examples."
>
> ```
> from NeuralCom.AutoCoNN import AutoCoNN
>
> INSTRUCT = 'Create an SOp that is the last letter of a word'
> VOCAB = ['a','b','c','d','e','f','g']
> EXAMPLE = [[['a','b','c'],['c','c','c']],[['b','d'],['d','d']]]
>
> auto = AutoCoNN()
> model,tokenizer = auto(instruct=INSTRUCT,vocab=VOCAB,example=EXAMPLE)
> ```
>
> The AutoCoNN process significantly reduces human effort while increasing scalability. Therefore, even for non-experts, the labor cost of authoring specialized models and authoring external tools will not differ too much.

---

> ### Author Response · Authors · 2023-11-15
> **Rebuttal by Authors: 2**
>
> > (2) Although this paper claims it does not require tool API, in my opinion, this is not free and has very large costs. If a calculator is there, I think we should use it instead of building neural networks mimicking these tools. API is also required for internet-based retrieval and data analysis (OpenAI Code Interpreter ), etc. I think adding a python api is simpler than adding a neural module.
>
> **We would like to stress that the Neural Comprehension framework is not intended to universally replace APIs but to offer a complementary and in many contexts a more suitable alternative—particularly for tasks that have smaller language model, demand end-to-end differentiability, or where invoking an external service might introduce latency issues.**
>
> First, we showd the performance of Vanilla CoT, PAL (a tool-based approach for Large language Models) and our NC method on the AddSub+ dataset in the original Figure 6. It covers mixed addition and subtraction Arithmetic Reasoning tasks with different numbers of digits from 1 to 20:
>
> **Table D-2:** Result of Figrue 6 for AddSub+ Dataset
> | Method | 1-dist AddSub | 5-dist AddSub | 10-dist AddSub | 15-dist AddSub | 20-dist AddSub |
> |-|-|-|-|-|-|
> | GPT-3.5+CoT | 81.7 | 65.3 | 28.6 | 8.9 | 3.2 |
> | GPT-3.5+PAL | **89.7** | 66.2 | 66.4 | 63.5 | 63.1 |
> | GPT-3.5+NC (Ours) | 81.9 | **70.2** | **67.5** | **67.8** | **67.4** |
> |-|-|-|-|-|-|
> | GLM-130B+CoT | **21.2** | 1.3 | 0.0 | 0.0 | 0.0 |
> | GLM-130B+PAL | 3.4 | 0.1 | 0.0 | 0.0 | 0.0 |
> | GLM-130B+NC (Ours) | **21.2** | **7.8** | **8.2** | **5.3** | **8.1** |
>
> To further demonstrate our performance, we selected the latest open-source models Llama-2-7B and Llama-2-70B and conducted comparative experiments using Vanilla CoT and PAL respectively as baseline methods on the Addition and Subtraction subset of GSM8K-Hard dataset:
>
> ---
>
> **Table D-3:** Result for Addition of GSM8K-hard Dataset
> | GSM8K-Hard (Addition) | llama-2-7b | llama-2-70b | GLM-130B |Avg |
> |-|-|-|-|-|
> |CoT| 6.3 | 43.8 | 6.3 | 18.8 |
> |PAL| **18.7** | 37.5 | 6.3 | 20.8 |
> |NC (Ours)| 12.5 | **43.8** | **7.2** | **21.2** |
>
> ---
>
> **Table D-4:** Result for Subtraction of GSM8K-hard Dataset
> | GSM8K-Hard (Subtraction) | llama-2-7b | llama-2-70b | GLM-130B |Avg |
> |-|-|-|-|-|
> |CoT| 6.3 | 27.9 | 1.8 | 12.0 |
> |PAL| 7.2 | 31.5 | 3.6 | 14.1 |
> |NC (Ours)| **9.9** | **32.4** | **4.5** | **15.6** |
>
> ---
>
> **Table D-5:** Result for Addition and Subtraction of GSM8K-hard Dataset
> | GSM8K-Hard (Addition and Subtraction ) | llama-2-7b | llama-2-70b | GLM-130B | Avg |
> |-|-|-|-|-|
> |CoT| 11.1 | 16.7 | 0.0 | 9.3 |
> |PAL| 11.1 | 27.8 | 5.6 | 14.8 |
> |NC (Ours)| **27.8** | **33.3** | **5.6** | **22.2** |
>
> ---
>
> We can see a universal conclusion: **for language models including Llama-2-7B, Llama-2-70B and GLM-130B, our method outperforms invoking calculators (PAL), and even GLM-130B invoking calculators on AddSub+ underperforms vanilla CoT. This shows that for current open-source models, external tools cannot be well utilized even when available. However, our method has unique advantages in that it no longer needs complex pre-processing to format external tool invocation or post-processing to interpret its output. Therefore it applies even to 60M-parameter T5 models (e.g. 27.3% -> 82.1% for LLC task in Figure 5).**
>
>
> We underscore that CoNNs, once compiled, are highly efficient at performing their designated tasks with minimal overhead. The AutoCoNN toolkit further mitigates the costs involved in generating new CoNNs by automating this process using existing language model capabilities, without the need to write extensive amounts of rule-based code from scratch.
>
> ---
>
> **We have resubmitted the supplementary file. The new version of the AutoCoNN toolkit allows CoNN to be implemented by Hugging Face (Transformers). All users can freely upload CoNN models generated by AutoCoNN, while users in need can directly download CoNN from Hugging Face and combine it with language models without recompiling the model.**
> ```
> # Upload CoNN
> model.push_to_hub("conn-model",organization="huggingface")
> tokenizer.push_to_hub("conn-model",organization="huggingface")
> ```
>
> ```
> # Download CoNN
> from NeuralCom.CoNN.modeling_conn import CoNNModel
> from NeuralCom.CoNN import Tokenizer
>
> model = CoNNModel.from_pretrained('conn-model')
> tokenizer = Tokenizer(model.config.input_encoding_map, model.config.output_encoding_map,model.config.max_position_embeddings)
> ```
>
> Therefore, we provide a low-cost way that does not require compilation, just as easy as downloading pre-trained language models. We will upload the CoNN mentioned in the paper, and we believe that with the growth of the community, there will be more and more CoNN models. We hope this can address your concerns.

---

> ### Author Response · Authors · 2023-11-15
> **Rebuttal by Authors: 3**
>
> > (3) Experimental results shown in Table 5 on real-world reasoning tasks, e.g. GSM8K, are similar to baselines without significant improvements.
>
> Responding to the concern about the efficacy on real-world reasoning tasks (e.g., GSM8K), we acknowledge that the results in Table 5 might lack the expected improvement. We aim to empower LMs with authentic symbolic reasoning capabilities without relying on external APIs, reducing dependency on implicit learning from textual data and ensuring robust performance on rule-intensive and out-of-distribution tasks.
>
> In fact, for the more challenging open-source GSM8K-Hard dataset [2] (addition and subtraction subsets), it brings an average 10% improvement for Llama-2-7B, Llama-2-70B and GLM-130B models. This is sufficient to demonstrate that introducing CoNN that enables precise rule reasoning can significantly enhance more difficult arithmetic reasoning tasks:
>
> **Table D-6:** Result for Addition and Subtraction of GSM8K-hard Dataset
> | **GSM8K-Hard (Addition and Subtraction )** | llama-2-7b | llama-2-70b | GLM-130B | Avg |
> |-|-|-|-|-|
> |CoT| 11.1 | 16.7 | 0.0 | 9.3 |
> |NC (Ours) | **27.8**(↑16.7) | **33.3**(↑16.6) | **5.6**(↑5.6) | **22.2**(↑12.9) |
>
> ---
>
> **The main goal of our work is to provide a method for robust symbolic operation and formal reasoning under the language model framework. When language models perform complex arithmetic reasoning, it may involve both reasoning and symbolic manipulation. Our method enables language models to have complete symbolic comprehension capability and fully solve symbolic operations problems. On this point, in recent years there are still many researchers conducting in-depth studies on language models performing simple symbolic operations [3-8].**
>
> ---
>
> We hope this information can help resolve your confusion and re-evaluate the rating of our paper. We appreciate the opportunity to refine our paper according to the reviewers' valuable recommendations.
>
> ---
>
> [1]Schick T, Dwivedi-Yu J, Dessì R, et al. Toolformer: Language models can teach themselves to use tools[J]. arXiv preprint arXiv:2302.04761, 2023.
>
> [2]PAL: Program-aided Language Models. Luyu Gao, Aman Madaan, Shuyan Zhou, Uri Alon, Pengfei Liu, Yiming Yang, Jamie Callan, Graham Neubig https://arxiv.org/abs/2211.10435
>
> [3] Nogueira, R., Jiang, Z., & Li, J.J. (2021). Investigating the Limitations of Transformers with Simple Arithmetic Tasks.
>
> [4]Qian, J., Wang, H., Li, Z., LI, S., & Yan, X. (2022). Limitations of Language Models in Arithmetic and Symbolic Induction. ArXiv, abs/2208.05051.
>
> [5]Cem Anil, Yuhuai Wu, Anders Johan Andreassen, Aitor Lewkowycz, Vedant Misra, Vinay Venkatesh Ramasesh, Ambrose Slone, Guy Gur-Ari, Ethan Dyer, and Behnam Neyshabur. Exploring length generalization in large language models. In Alice H. Oh, Alekh Agarwal, Danielle Belgrave, and Kyunghyun Cho (eds.), Advances in Neural Information Processing Systems, 2022.
>
> [6]Hattie Zhou, Azade Nova, Hugo Larochelle, Aaron Courville, Behnam Neyshabur, and Hanie Sedghi. Teaching algorithmic reasoning via in-context learning. arXiv preprint arXiv:2211.09066, 2022b.
>
> [7] Lee, N., Sreenivasan, K.K., Lee, J.D., Lee, K., & Papailiopoulos, D. (2023). Teaching Arithmetic to Small Transformers. ArXiv, abs/2307.03381.
>
> [8]Xu, X., Pan, Z., Zhang, H., & Yang, Y. (2023). It Ain't That Bad: Understanding the Mysterious Performance Drop in OOD Generalization for Generative Transformer Models. ArXiv, abs/2308.08268.

---

### Official Review · Reviewer_M9Rq · 2023-10-31

**Soundness:** 3 good
**Presentation:** 2 fair
**Contribution:** 2 fair
**Rating:** 6
**Confidence:** 3

**Summary:**

This work introduces Neural Comprehension, a method for combining language models with compiled neural networks (CoNNs) which allow explicit rules to be encoded as transformers. CoNNs can generalize more effectively on symbolic problems, but do not have the linguistic flexibility of LMs. My understanding is that Neural Comprehension works "similar to MoE" (mixture of experts), in that the token at a given position is decoded either by the LM or by a CoNN, depending on whether explicit reasoning is needed. The authors demonstrate that Neural Comprehension indeed generalizes better than a pure LM, especially for problems that might have very long length (e.g Figure 4).

**Strengths:**

- LMs often do not generalize well in symbolic reasoning problems, so finding ways to improve this is very important
- experiments are quite extensive
- Neural Comprehension seems to work reasonably well on many of the tasks the authors look at

**Weaknesses:**

- Presentation is overall quite hard to follow. Authors should correct me if I misinterpret any parts of the method below
- It is not clear when aspects of CoNNs are learned or if they are just compiled as fully operational symbolic functions. Gradients are mentioned, but the authors do not elaborate on exactly what these apply to
- The combination of CoNNs and LMs seems quite simplistic here based on Figure 3 and section 4.1. For example, it seems that Neural Comprehension takes the output of the Addition CoNN when encountering '='. It is not clear whether there is a more sophisticated process used to decide when to apply CoNNs. If not, why is using CoNNs better than simply using external tools?
- In a similar vein, it would strengthen the results to also compare to networks which use external tools. The authors frame the work in contrast to that, but it is not clear how those methods would compare

**Questions:**

See weaknesses

---

> ### Author Response · Authors · 2023-11-15
> **Rebuttal by Authors:  1**
>
> **Thank you for recognizing the significance of our work in addressing generalization challenges associated with symbolic reasoning tasks in language models (LMs). We appreciate the opportunity to clarify Neural Comprehension and its contribution to the field.**
>
> > It is not clear when aspects of CoNNs are learned or if they are just compiled as fully operational symbolic functions. Gradients are mentioned, but the authors do not elaborate on exactly what these apply to
>
> 1. Regarding the learning aspects of CoNNs: CoNNs are intentionally designed with explicit rules and are not meant to be learned in the traditional sense. Their parameters are crafted based on predefined symbolic functions and do not necessitate further gradient-based learning. This design choice ensures that CoNNs can perform their specified symbolic reasoning tasks with flawless accuracy, leveraging the attention mechanism intrinsic to transformer model. All CoNNs use standard Transformer architectures, so they are easy to incorporate into existing language models. Table 2 in the appendix shows the specific architectures for all CoNNs. For example, the Parity model is a 4 layer, 132 hidden size Transformer with 2.2M parameters.
>
> 2. Gradient mentions in our work relate to the integration of CoNNs within the LM framework. We presented a novel gating mechanism that facilitates the model's decision to transition between LM and CoNN processing based on \(\beta\), which is pre-determined during the construction of each CoNN. This piecewise function perspective allows us to controllably influence the gradient updates, optimizing the coupling of implicit learning via LM and explicit rule execution via CoNNs. The gradient updates from CoNNs provide the model with an immediate and optimal adjustment specific to rule-based tasks.
>
> 3. The role of gradients in our work is, therefore, to enhance the in-context learning capacity of LMs when coupled with rule-based knowledge from CoNNs. By incorporating CoNN modules directly into the architecture of LMs, we aim to endow the resulting model with genuine symbolic comprehension capabilities that transcend the limitations of LMs when confronted with out-of-distribution and rule-intensive tasks.

---

> ### Author Response · Authors · 2023-11-15
> **Rebuttal by Authors: 2**
>
> > The combination of CoNNs and LMs seems quite simplistic here based on Figure 3 and section 4.1. For example, it seems that Neural Comprehension takes the output of the Addition CoNN when encountering '='. It is not clear whether there is a more sophisticated process used to decide when to apply CoNNs. If not, why is using CoNNs better than simply using external tools?
>
> The gated mechanism introduced in Neural Comprehension indeed uses a predefined trigger based on the '=' symbol as an initial demonstration of the concept. This plug-and-play approach makes it easy to incorporate CoNNs into language model frameworks that also use Transformer architectures. On the other hand, in Figure 7 of the main text we show the effect of combining multiple CoNNs together. For example, the AddSub task requires using both the Addition Model and Subtraction Model, as well as mixing in other uncorrelated CoNNs. The experimental results show that this simple combination mechanism is still able to handle the work of every CoNN quite well:
>
>
> **Table C-1:** Result of Figrue 7 for 20-Dists of AddSub+
> | GSM8K-Hard (Subtraction) | Vanilla CoT | Add | Sub | (Uncorrelated) Parity | (Uncorrelated) Laster-Word |  (Uncorrelated) Copy-Letter |
> |-|-|-|-|-|-|-|
> |GPT-3.5| 6.9 | 44.9 | 64.4 | 64.4 | 64.4 | 64.4 |
> |-|-|-|-|-|-|-|
> |GPT-3| 0 | 19.5 | 36.9 | 36.9 | 36.9 | 36.9 |
> |-|-|-|-|-|-|-|
> |GLM-130B| 0 | 5.5 | 7.6 | 7.6 | 7.6 | 7.6 | 7.6 |
>
> **In Table C-1 from left to right we gradually add new CoNNs.**
>
> Add and Sub are needed for the mixed Arithmetic reasoning task, so they can further improve over Vanilla CoT for all three models. On the other hand, adding irrelevant CoNNs does not degrade our performance, **which shows our method is able to balance when to apply multiple CoNNs.**
>
> ---
>
> The use of CoNNs is fundamentally different from external tools as it maintains a fully neural structure within the LM framework, ensuring end-to-end differentiability and unified neural architecture. This design choice eliminates the need for generating additional code or relying on an external interpreter, thus improving efficiency and directness in operations. Compared to using external tools, Neural Comprehension shows advantage in two aspects:
>
> 1. Generation speed: Neural Comprehension can increase most half generation speed compared to Call-Function (Because there is no need to generate additional code).
>
> 2. Independent on model generation ability: Neural Comprehension can achieve superior symbolic computing performance on both small and large models, while call external tools depend on the generation code ability of the model, which have restrictions.
>
> Here is a simple arithmetic example: "Alex's fitness tracker shows he ran 12285 meters this week, and ran 9221 meters last week. So how many more meters did he run this week?" For models that can directly utilize external tools, our method can forward infer without interruption to access the external tools, speeding up inference without compromising performance.
>
> ---
>
> ### External Tools (Call Function for ChatGPT)
>
> Alex ran 12285 meters this week, and he ran 9221 meters last week. So the difference is: 12285 - 9221 = `<FUNCTION>{'name':'CALL_PYTHON','arguments':'def sub():\n return 12285 - 9221\nsub()'}</FUNCTION>[EOS]<RETURN>{'name':'CALL_PYTHON','content':'3064'}</RETURN><EOS>` Therefore, Alex ran 3064 more meters this week compared to last week.
>
> ### Neural Comprehension (Ours)
>
> Alex ran 12285 meters this week, and he ran 9221 meters last week. So the difference is: 12285 - 9221 = *3064*. Therefore, Alex ran 3064 more meters this week compared to last week.

---

> ### Author Response · Authors · 2023-11-15
> **Rebuttal by Authors: 3**
>
> > In a similar vein, it would strengthen the results to also compare to networks which use external tools. The authors frame the work in contrast to that, but it is not clear how those methods would compare
>
> We appreciate the reviewer's suggestion to include comparisons with methods utilizing external tools. **In Figure 6 of section 5.3 in the main text, we have already compared our method with PAL [1] (Program-Aided Language Models, it converts the task solving process into Python code, and invokes Python to execute the code to produce the final answer.) on the Arithmetic reasoning task (AddSub+)**, which encompasses 1-digit to 20-digit addition and subtraction arithmetic reasoning. We excerpt some of the experimental results in the table below:
>
> **Table C-2:** Result of Figrue 6 for AddSub+ Dataset
> | Method | 1-dist AddSub | 5-dist AddSub | 10-dist AddSub | 15-dist AddSub | 20-dist AddSub |
> |-|-|-|-|-|-|
> | GPT-3.5+CoT | 81.7 | 65.3 | 28.6 | 8.9 | 3.2 |
> | GPT-3.5+PAL | **89.7** | 66.2 | 66.4 | 63.5 | 63.1 |
> | GPT-3.5+NC (Ours) | 81.9 | **70.2** | **67.5** | **67.8** | **67.4** |
> |-|-|-|-|-|-|
> | GLM-130B+CoT | **21.2** | 1.3 | 0.0 | 0.0 | 0.0 |
> | GLM-130B+PAL | 3.4 | 0.1 | 0.0 | 0.0 | 0.0 |
> | GLM-130B+NC (Ours) | **21.2** | **7.8** | **8.2** | **5.3** | **8.1** |
>
>
> We further supplemented experiments on the addition and subtraction subsets of GSM8K-Hard [1]. We chose the open-source Llama-2-7B, Llama-2-70B and GLM-130B models, and took Vanilla CoT and PAL as baselines respectively:
>
> **Table C-3:** Result for Addition of GSM8K-hard Dataset
> | GSM8K-Hard (Addition) | llama-2-7b | llama-2-70b | GLM-130B |Avg |
> |-|-|-|-|-|
> |CoT| 6.3 | 43.8 | 6.3 | 18.8 |
> |PAL| **18.7** | 37.5 | 6.3 | 20.8 |
> |NC (Ours)| 12.5 | **43.8** | **7.2** | **21.2** |
>
> ---
>
> **Table C-4:** Result for Subtraction of GSM8K-hard Dataset
> | GSM8K-Hard (Subtraction) | llama-2-7b | llama-2-70b | GLM-130B |Avg |
> |-|-|-|-|-|
> |CoT| 6.3 | 27.9 | 1.8 | 12.0 |
> |PAL| 7.2 | 31.5 | 3.6 | 14.1 |
> |NC (Ours)| **9.9** | **32.4** | **4.5** | **15.6** |
>
> ---
>
> **Table C-5:** Result for Addition and Subtraction of GSM8K-hard Dataset
> | GSM8K-Hard (Addition and Subtraction ) | llama-2-7b | llama-2-70b | GLM-130B | Avg |
> |-|-|-|-|-|
> |CoT| 11.1 | 16.7 | 0.0 | 9.3 |
> |PAL| 11.1 | 27.8 | 5.6 | 14.8 |
> |NC (Ours)| **27.8** | **33.3** | **5.6** | **22.2** |
>
> ---
>
> **Experiments prove that our method NC performs better than the CoT and PAL on all three datasets. One major reason is that invoking external tools requires generating accurate code, which may be error-prone or irrelevant for many language models. But our method is plug-and-play and does not rely on the language model's ability to generate code, thus even applies to 60M-parameter T5-small models (For example in Figure 5, T5-Small improves from 27.3% to 82.1% on Last Letter Concatenation.).**
>
> ---
>
> We hope these revisions and expansions address the reviewer's concerns and can re-evaluate the rating of our paper. We thank the reviewer for the constructive feedback which has greatly helped refine our work.
>
> ---
>
> [1]PAL: Program-aided Language Models. Luyu Gao, Aman Madaan, Shuyan Zhou, Uri Alon, Pengfei Liu, Yiming Yang, Jamie Callan, Graham Neubig https://arxiv.org/abs/2211.10435

---

> > ### Comment · Reviewer_M9Rq · 2023-11-23
> >
> > While I still have some concerns about generality of the approach (e.g. requiring a user-defined trigger) and other concerns raised by reviewers (difficulty in expanding to many tasks, issues with cost etc.), the authors do clarify many of my questions and issues. I will raise my score.

---

> ### Author Response · Authors · 2023-11-23
>
> Thank you for your follow-up comments and for acknowledging the clarifications provided in our response. We aim to address your remaining concerns about the generality of the approach and other issues raised.
>
>
> We understand the importance of scalability and applicability to a vast range of tasks. Our AutoCoNN toolkit was introduced to specifically address this challenge by enabling efficient and simplified generation of CoNNs for new tasks. This development significantly streamlines the process of expanding the model’s capabilities and reduces the manual effort required.  AutoCoNN has high accuracy. The Parity Model, Reverse Model, Copy Model and Last-letter Model involved in this paper were all generated by AutoCoNN.
>
> We acknowledge that incorporating specialized neural modules comes with its own set of challenges, including potential costs associated with their development and maintenance. However, once developed, these CoNNs are reusable across different tasks and models, amortizing the initial investment over time. Additionally, the high efficiency and accuracy of CoNNs can lead to a reduction in computational costs during deployment, as they streamline the process of rule-based reasoning.
>
> **Therefore, we have revised the Supplementary Material. The new version of the AutoCoNN toolkit allows CoNN to be implemented by Hugging Face (Transformers). On the one hand, all users can freely upload CoNN models generated by AutoCoNN just like pretrained language models in the past. On the other hand, for users in need, they can directly download CoNN from Hugging Face and combine it with language models without rebuilding. This process simplifies the cost of using CoNN and increases its scalability.**
>
> > Upload CoNN:
> ```python
> from NeuralCom.AutoCoNN import AutoCoNN
>
> INSTRUCT = 'Create an SOp that is the last letter of a word'
> VOCAB = ['a','b','c','d','e','f','g']
> EXAMPLE = [[['a','b','c'],['c','c','c']],[['b','d'],['d','d']]]
>
> auto = AutoCoNN()
> model,tokenizer = auto(instruct=INSTRUCT,vocab=VOCAB,example=EXAMPLE)
>
> model.push_to_hub("conn-model",organization="huggingface")
> tokenizer.push_to_hub("conn-model",organization="huggingface")
> ```
>
> > Download CoNN:
> ```python
> from NeuralCom.CoNN.modeling_conn import CoNNModel
> from NeuralCom.CoNN import Tokenizer
>
>
> model = CoNNModel.from_pretrained('conn-model')
> tokenizer = Tokenizer(model.config.input_encoding_map, model.config.output_encoding_map,model.config.max_position_embeddings)
>
> ```
>
>
>
> We expect that, through ongoing research and community contributions, the catalog of readily available CoNNs will grow, thereby enhancing the model's versatility and reach.
>
> ---
>
> **We appreciate the opportunity to clarify these points and are grateful for the change in score as a reflection of your updated consideration.  Thank you once again for your constructive suggestions and support!**

---

### Official Review · Reviewer_68he · 2023-11-01

**Soundness:** 3 good
**Presentation:** 2 fair
**Contribution:** 3 good
**Rating:** 5
**Confidence:** 3

**Summary:**

This paper addresses the limitations of language models (LMs) in handling deterministic symbolic reasoning and rule-based tasks due to their implicit learning from textual data. To enhance their rule comprehension ability, the authors introduce "Neural Comprehension," which incorporates Compiled Neural Networks (CoNNs) into LMs. CoNNs are transformer-based neural networks that execute rules using artificially generated attention weights. This method enables LMs to effectively tackle rule-intensive challenges and improves their performance in symbolic reasoning tasks and real-world calculations.

**Strengths:**

1. The paper introduces a novel approach, "Neural Comprehension" by incorporating Compiled Neural Networks (CoNNs) into language models. This approach is highly original as it addresses the limitations of LMs in handling deterministic symbolic reasoning, offering an innovative solution that combines neural networks and explicitly coded, interpretable CoNNs;

2. The research methodology is well-founded, with empirical experiments showing superior performance compared to existing techniques. The study also includes a thorough analysis of combining multiple CoNNs, offering insights into model fusion. Besides, the AutoCoNN toolkit for automatic CoNN generation adds to the quality and scalability of the approach.

**Weaknesses:**

1. Even though the use of CoNNs to handle symbolic reasoning tasks is meticulously explained, the authors lack a well-structured explanation of "Neural Comprehension" and CoNNs, making it not very accessible to readers;

2. As I understand it, the authors need to generate a corresponding CoNN for each rule-based task (such as Parity, Reverse, Subtraction, and so on) and integrate it with the LM, is that correct? If so, this approach seems to lack generality.

**Questions:**

1. What does $I_{d_{L}}$ in Equation 1 mean? The author seems to have not provided an explanation. This seriously impedes my understanding of the crucial part of the proposed method;

2. What does "ICL" mean? Which term is the abbreviation for?

3. How are CoNNs compiled? How is the gradient $I_{d_{2}}$ for rule-based in Equation 2 obtained? Does it involve first training a specific task's CoNNs and then extracting the rule-based gradient, which is merged with the LM's gradient to influence the final gradient? If so, how are CoNNs trained? What is the training dataset like?

4. From all the equations, I haven't seen an explanation of how the LM-based gradient and Rule-based gradient are combined. I feel very confused about this and need a more detailed explanation from the authors.

Typos:

1. ’=’  ->  ‘=’，  ’Select’  ->  ‘Select’

2. “Neuralresulting”

3. As depicted in Figure 5.4  ->  As depicted in Figure 7

---

> ### Author Response · Authors · 2023-11-15
> **Rebuttal by Authors: 1**
>
> Thank you for your insightful comments and valuable questions. We provide necessary clarifications below and have highlighted more improvement.
>
> >  Even though the use of CoNNs to handle symbolic reasoning tasks is meticulously explained, the authors **lack a well-structured explanation of "Neural Comprehension" and CoNNs**, making it not very accessible to readers;
>
>
> **CoNN** is a standard transformer structure network, for example the Parity model is a 4 layer, 132 hidden size Transformer with 2.2M parameters. They are specially designed neural networks with weights and structures directly encoding the symbolic rules, allowing for deterministic execution of rule-based tasks.
>
> **Neural Comprehension** is a MoE-style neural network framework we designed, which is entirely composed of neural networks and maintains the generative seq2seq style. It uses predefined CoNNs as gating to determine when to output results from CoNNs. This approach is simple and plug-and-play, allowing combination with pretrained language models.
>
> To clarify "Neural Comprehension" and CoNNs, We have **reconstructed this paper's sections to describe these concepts more vividly, so they are accessible to all readers.** For more detailed clarification of CoNN and Comprehension, please see Rebuttal by Authors: 2 (https://openreview.net/forum?id=9nsNyN0vox&noteId=fqxtXeLATK)
>
> >  As I understand it, the authors need to generate a corresponding CoNN for each rule-based task (such as Parity, Reverse, Subtraction, and so on) and integrate it with the LM, is that correct? If so, this approach seems to lack generality.
>
> We understand the concerns about CoNNs’ generality, to address this, we also proposed the AutoCoNN toolkit (4.3), which streamlines the creation of CoNNs for new tasks by leveraging instructive cues from pre-trained LMs. We provide 24 different examples that are used to generate CoNNs. This process can easily generalize to tasks not covered by existing CoNNs. Which can significantly extend the generality and application scope of our method Neural Comprehension. We also provide simple code of AutoCoNN for rapid creation of new CoNNs :
>
> ```
> from NeuralCom.AutoCoNN import AutoCoNN
>
> INSTRUCT = 'Create an SOp that is the last letter of a word'
> VOCAB = ['a','b','c','d','e','f','g']
> EXAMPLE = [[['a','b','c'],['c','c','c']],[['b','d'],['d','d']]]
>
> auto = AutoCoNN()
> model,tokenizer = auto(instruct=INSTRUCT,vocab=VOCAB,example=EXAMPLE)
> ```
>
> ---
>
> On the other hand, we have modified the Supplementary Material. The new version of the AutoCoNN toolkit allows CoNN to be implemented by Hugging Face (Transformers). All users can freely upload the CoNN models generated by AutoCoNN, while users in need can directly download CoNN from Hugging Face and combine it with language models without recompiling the model. We believe that as the community grows, more and more CoNNs will be compiled and uploaded. We hope this can address your concerns about generality and cost.
>
> ```
> # Upload CoNN
> model.push_to_hub("conn-model",organization="huggingface")
> tokenizer.push_to_hub("conn-model",organization="huggingface")
> ```
>
> ```
> # Download CoNN
> from NeuralCom.CoNN.modeling_conn import CoNNModel
> from NeuralCom.CoNN import Tokenizer
>
> model = CoNNModel.from_pretrained('conn-model')
> tokenizer = Tokenizer(model.config.input_encoding_map, model.config.output_encoding_map,model.config.max_position_embeddings)
> ```
>
> **Therefore, AutoCoNN can be easily and conveniently extended. Even compared to manually writing tools, it does not increase too much labor cost.**

---

> ### Author Response · Authors · 2023-11-15
> **Rebuttal by Authors: 2**
>
> ### Question
>
> >  What does $I_{d_L}$ in Equation 1 mean? The author seems to have not provided an explanation. This seriously impedes my understanding of the crucial part of the proposed method;
>
> In Equation 1, $I_{d_L}$ represents the identity matrix corresponding to the dimension of the LM's vocabulary size, where shown in page 4’ footnote `using identity matrices $I_{d_L}$ (for $d_L$) and $I_{d_C}$ (for $d_C$ ) to concatenate them for model fusion.`. It serves to preserve the LM's original output distribution for the predicted token when the gating mechanism opts to use the LM's output ($\beta=0$). For scenarios requiring rule-based computation ($\beta=1$), the CoNN's output, $H_C$, from a specialized subset of the token vocabulary is considered. We will ensure all notations are thoroughly explained in the revised paper.
>
> To improve readability, we have reorganized this footnote content into the Method section to enhance clarity.
>
> >  What does "ICL" mean? Which term is the abbreviation for?
>
> ICL stands for "In-Context Learning," where a model adapts its outputs based on context provided directly within its input sequence.
>
> >  How are CoNNs compiled? How is the gradient $I_{d_2}$ for rule-based in Equation 2 obtained? Does it involve first training a specific task's CoNNs and then extracting the rule-based gradient, which is merged with the LM's gradient to influence the final gradient? If so, how are CoNNs trained? What is the training dataset like?
>
>
> The weights and operations in CoNN are fixed and do not require training, as they are predetermined to perform their task correctly by construction. Figure 2 in the main text showcases an example of a CoNN designed for the Parity task. Each operational step within the CoNN corresponds to a direct transformation similar to mutual transformations between layers in transformers – the Select operation simulates attention over certain positions in the input sequence, Aggregate gathers and combines information across positions, and Zipmap conducts token-level computations corresponding to feed-forward layers in the transformer. The compilation process turns the abstract computational steps into tangible weight matrices in the transformer model, which embody the logic of the function or rule.
>
> For the AutoCoNN toolkit, it generates Tracr [1] code that describes sequence operation processes based on directives and examples. After that, the code is compiled into Pytorch-style transformer weights and loads them.
>
> The Equation 2 represents gradient combination process of CoNNs and large language models. $I_{d_2}$ signifies the CoNN's deterministic gradient derived from the explicit rules it implements, contrasting with $I_{d_1}$, which is the gradient learned by the LM from data. When a CoNN processes an input to produce an output, these rule-based operations occur and propagate gradients back through the explicit transformation mappings they represent. The content of this paragraph is to prove from the gradient perspective that for language models, incorporating CoNN can have a positive impact.
>
> It is worth noting that the meaning of $I_{d_1}$ and $I_{d_2}$ here are not identity matrix but gradient. For clarity in the paper, we have changed $I_{d_1}$ and $I_{d_2}$ in the main text to $G_T$ and $G_R$.

---

> ### Author Response · Authors · 2023-11-15
> **Rebuttal by Authors: 3**
>
> > From all the equations, I haven't seen an explanation of how the LM-based gradient and Rule-based gradient are combined. I feel very confused about this and need a more detailed explanation from the authors.
>
> **The Section 4.2 provides an explanation of Section 4.1 from the perspective of In-Context Learning, with the goal of illustrating that by combining CoNNs, it is possible to optimize the generated content of rule-based texts during the generation process.**
>
> In our proposed Neural Comprehension framework, we employ a gating mechanism to control which gradient – either from the LM or CoNN – is applied for each token generation step during the in-context learning process. This choice is dictated by the presence of a rule-trigger context (as indicated by our defined gating variable $\beta$). When $\beta = 1$, indicating a rule-intensive computation, the gradient from CoNN ($I_{d_2}$) is prioritized, effectively overwriting the LM's gradient ($I_{d_1}$) for that particular computation.
>
> However, it is crucial to note that this does not imply a physical combination or averaging of the two distinct gradients. Instead, we make CoNNs “take charge” of rule-based transformations, ensuring precision and correctness in such tasks, with LMs managing the broader language understanding.
>
> Moreover, Equation 2 sets forth the separation between text-derived and rule-derived gradients. The aforementioned gating mechanism – informed by the presence of a rule/task-specific context – determines the utilization of gradients for generating each token. The gating control has been intentionally simplified to preserve transparency and simplicity, enabling an easy understanding of when and why rule-based computations are executed.
>
> > Correction of Typos
>
> We appreciate you pointing out the typos. We have submitted a revised manuscript , which have corrected these typos and ensured a thoroughly proofread and clearer revision about above issues.
>
> ---
>
> **We believe these measures will substantially enhance the paper's clarity and presentation, satisfactorily addressing the concerns raised. Thank you again for your valuable feedback, and we would very much welcome further discussion. We look forward to you re-evaluating our work.**
>
> ---
>
> [1] Lindner, D., Kram'ar, J., Rahtz, M., McGrath, T., & Mikulik, V. (2023). Tracr: Compiled Transformers as a Laboratory for Interpretability. ArXiv, abs/2301.05062.

---

### Official Review · Reviewer_NkHa · 2023-11-01

**Soundness:** 3 good
**Presentation:** 3 good
**Contribution:** 3 good
**Rating:** 8
**Confidence:** 4

**Summary:**

- This paper proposes a new method called "Neural Comprehension" to enhance the symbolic reasoning capabilities of LLMs by integrating compiled neural networks (CoNNs), which are transformer-based networks with artificially designed attention weights to explicitly implement rules and transformations, thus can accurately perform symbolic computations.
- Neural Comprehension outperforms baselines like fine-tuning and few-shot learning on length generalization, and achieves near perfect accuracy on symbolic operations like parity, reverse, addition, and subtraction.

**Strengths:**

- The gating mechanism naturally incorporates CoNN into LLMs and allows gradient optimization for the whole network. This feature makes the proposed method different from the existing tool-use papers.
- Near perfect accuracy on symbolic tasks, significant gains on arithmetic reasoning, compared to the regular LLMs like GPT-3/3.5, especially when the number of digits goes up.
- The proposed method results in less than 3% increase in the inference time, showing the potential to be applied to the existing models efficiently.

**Weaknesses:**

- Marginal performance improvement on real-world arithmetic reasoning tasks such as GSM8K. The result suggests that it may not be necessary for us to apply such neural compression into LLMs if the task is not related to strict symbolic operations.

**Questions:**

- In table 5 of appendix, it seems that you plug the neural comprehension into GPT-3. How did you achieve that when GPT-3 can only be accessed through APIs?
- Have you ever tried the [GSM-hard](https://huggingface.co/datasets/reasoning-machines/gsm-hard) dataset? [1] I think the proposed method is helpful for problems that have more digits. I expect that this method will be useful for GSM-hard as it is intended to includes more digits in the answers. Can you try experiments on it and compare the results with PAL?



[1] PAL: Program-aided Language Models. Luyu Gao, Aman Madaan, Shuyan Zhou, Uri Alon, Pengfei Liu, Yiming Yang, Jamie Callan, Graham Neubig https://arxiv.org/abs/2211.10435

---

> ### Author Response · Authors · 2023-11-15
> **Rebuttal by Authors: 1**
>
> **We are grateful for the insightful feedback and constructive critique provided by the reviewers. We have carefully considered each point raised and provide our responses below:**
>
> > Marginal performance improvement on real-world arithmetic reasoning tasks such as GSM8K. The result suggests that it may not be necessary for us to apply such neural compression into LLMs if the task is not related to strict symbolic operations.
>
> Regarding the marginal improvement on real-world arithmetic reasoning tasks such as GSM8K, we appreciate your perspective. We acknowledge that the most substantial benefits of "Neural Comprehension" (NC) are observed in tasks with strict symbolic rule requirements. Nonetheless, we argue that even moderate enhancement in arithmetic reasoning demonstrates the viability of integrating explicit rule-based computation within LLMs for a broader application scope, which warrants further exploration.
>
> **In fact, the main contribution of our work is not just the actual real-world application. As the title suggests, our method allows language models to master symbolic operations themselves for the first time. We explored a robust formal reasoning approach that can ensure language models perform symbolic tasks stably without being limited by task length and complexity.**
>
> In recent years, many researchers are still conducting in-depth studies on how language models (including large language models) can solve simple, toy symbolic operation tasks [2-7]. **Our work solves this problem and also delivers competitive performance in arithmetic reasoning tasks (e.g. compared to PAL on GSM8K-Hard and AddSub+ datasets below).**
>
>
> > In table 5 of appendix, it seems that you plug the neural comprehension into GPT-3. How did you achieve that when GPT-3 can only be accessed through APIs?
>
> To compare with CoT and PAL [1] which experimented on GPT-3 series models, we simulated Neural Comprehension (NC) within the constraints of the API access. We treated the API's output as if it were part of the Neural Comprehension structure. This involved a simulated gating mechanism, where the output from the API was corrected using CoNNs form left to right, and then the adjusted response (Truncate the text after the altered text.) was changed into the API' input for continued generation. This simulation was to ensure that the benefits of NC could be compared fairly with the existing results from PAL.

---

> ### Author Response · Authors · 2023-11-15
> **Rebuttal by Authors: 2**
>
> > Have you ever tried the GSM-hard dataset? [1] I think the proposed method is helpful for problems that have more digits. I expect that this method will be useful for GSM-hard as it is intended to includes more digits in the answers. Can you try experiments on it and compare the results with PAL?
>
> On one hand, Figure 6 in the main text shows comparison experiments with CoT and PAL on the AddSub+ dataset, which involves mixed 1 to 20 digit addition and subtraction, with 400 samples for each digit, compared to GSM-hard it can demonstrate reasoning abilities involving different number of digits:
>
>
> **Table A-1:** Result of Figrue 6 for AddSub+ Dataset
> | Method | 1-dist AddSub | 5-dist AddSub | 10-dist AddSub | 15-dist AddSub | 20-dist AddSub |
> |-|-|-|-|-|-|
> | GPT-3.5+CoT | 81.7 | 65.3 | 28.6 | 8.9 | 3.2 |
> | GPT-3.5+PAL | **89.7** | 66.2 | 66.4 | 63.5 | 63.1 |
> | GPT-3.5+NC (Ours) | 81.9 | **70.2** | **67.5** | **67.8** | **67.4** |
> |-|-|-|-|-|-|
> | GLM-130B+CoT | **21.2** | 1.3 | 0.0 | 0.0 | 0.0 |
> | GLM-130B+PAL | 3.4 | 0.1 | 0.0 | 0.0 | 0.0 |
> | GLM-130B+NC (Ours) | **21.2** | **7.8** | **8.2** | **5.3** | **8.1** |
>
>
> On other hand, we appreciate the suggestion to evaluate our method on the GSM-hard dataset. **We conducted additional experiments on the open-source Llama-2-7B, Llama-2-70B and GLM-130B models using both Vanilla CoT and PAL as baselines**, on the Addition and Subtraction subsets of the GSM-Hard dataset. The results indicate that Neural Comprehension enhances the models' performance on arithmetic reasoning tasks with large digit counts, showing a consistent increase in solve rates for both addition and subtraction tasks, outperforming both CoT and PAL on the combined Addition and Subtraction scores:
>
> ---
>
> **Table A-2:** Result for Addition of GSM8K-hard Dataset
> | GSM8K-Hard (Addition) | llama-2-7b | llama-2-70b | GLM-130B |Avg |
> |-|-|-|-|-|
> |CoT| 6.3 | 43.8 | 6.3 | 18.8 |
> |PAL| **18.7** | 37.5 | 6.3 | 20.8 |
> |NC (Ours)| 12.5 | **43.8** | **7.2** | **21.2** |
>
> ---
>
> **Table A-3:** Result for Subtraction of GSM8K-hard Dataset
> | GSM8K-Hard (Subtraction) | llama-2-7b | llama-2-70b | GLM-130B |Avg |
> |-|-|-|-|-|
> |CoT| 6.3 | 27.9 | 1.8 | 12.0 |
> |PAL| 7.2 | 31.5 | 3.6 | 14.1 |
> |NC (Ours)| **9.9** | **32.4** | **4.5** | **15.6** |
>
>
> ---
>
> **Table A-4:** Result for Addition and Subtraction of GSM8K-hard Dataset
> | GSM8K-Hard (Addition and Subtraction ) | llama-2-7b | llama-2-70b | GLM-130B | Avg |
> |-|-|-|-|-|
> |CoT| 11.1 | 16.7 | 0.0 | 9.3 |
> |PAL| 11.1 | 27.8 | 5.6 | 14.8 |
> |NC (Ours)| **27.8** | **33.3** | **5.6** | **22.2** |
>
> ---
>
> One major reason is that for these open-sourced models, they may not necessarily have very good code generation capabilities, and cannot accurately generate code to invoke APIs and process the results. **But our work adapts existing language models in a plug-and-play manner, without needing external scaffolding tools, and can even be applied on a 60M T5-Small model (in Figure 5)**.
>
> We believe that our Neural Comprehension approach demonstrates substantial potential for advancing the state-of-the-art in symbolic reasoning for language models and are excited about its possibilities. Thank you again for your valuable feedback, which will guide our work towards improved clarity and further experimentation.
>
> ---
>
> [1]PAL: Program-aided Language Models. Luyu Gao, Aman Madaan, Shuyan Zhou, Uri Alon, Pengfei Liu, Yiming Yang, Jamie Callan, Graham Neubig https://arxiv.org/abs/2211.10435
>
> [2]Nogueira, R., Jiang, Z., & Li, J.J. (2021). Investigating the Limitations of Transformers with Simple Arithmetic Tasks.
>
> [3]Qian, J., Wang, H., Li, Z., LI, S., & Yan, X. (2022). Limitations of Language Models in Arithmetic and Symbolic Induction. ArXiv, abs/2208.05051.
>
> [4]Cem Anil, Yuhuai Wu, Anders Johan Andreassen, Aitor Lewkowycz, Vedant Misra, Vinay Venkatesh Ramasesh, Ambrose Slone, Guy Gur-Ari, Ethan Dyer, and Behnam Neyshabur. Exploring length generalization in large language models. In Alice H. Oh, Alekh Agarwal, Danielle Belgrave, and Kyunghyun Cho (eds.), Advances in Neural Information Processing Systems, 2022.
>
> [5]Hattie Zhou, Azade Nova, Hugo Larochelle, Aaron Courville, Behnam Neyshabur, and Hanie Sedghi. Teaching algorithmic reasoning via in-context learning. arXiv preprint arXiv:2211.09066, 2022b.
>
> [6] Lee, N., Sreenivasan, K.K., Lee, J.D., Lee, K., & Papailiopoulos, D. (2023). Teaching Arithmetic to Small Transformers. ArXiv, abs/2307.03381.
>
> [7]Xu, X., Pan, Z., Zhang, H., & Yang, Y. (2023). It Ain't That Bad: Understanding the Mysterious Performance Drop in OOD Generalization for Generative Transformer Models. ArXiv, abs/2308.08268.

---

> ### Comment · Reviewer_NkHa · 2023-11-21
> **Thanks for the response**
>
> Thanks for the additional experiments to address my questions. I am satisfied with the current results that consistently demonstrate the effectiveness of long-digit operations. I am willing to raise my score to 8.

---

> > ### Author Response · Authors · 2023-11-22
> >
> > **We deeply appreciate you taking the time to thoroughly review ours work and provide thoughtful feedback. Raising your score to 8 is incredibly encouraging. We are pleased the additional experiments adequately addressed your questions. Thank you for your openness and willingness to reconsider based on new evidence. Your flexible and fair review process facilitates quality research!**

---

### Author Response · Authors · 2023-11-15
**Overall Response to Reviews**

**We appreciate all the reviewers for their constructive feedback and recognition of the contributions of our paper!**

---

### Strengths:

1. **Novelty**: Reviewers NkHa and 68he appreciate the innovative "Neural Comprehension" approach that incorporates Compiled Neural Networks (CoNNs) into language models to address limitations in deterministic symbolic reasoning. This method represents a significant original contribution to enhancing rule comprehension ability within LMs.

2. **Sound Methodology and Strong Performance**: Reviewers 68he and uMuv note the well-founded research methodology, with empirical evidence of superior performance in symbolic reasoning tasks and real-world calculations compared to existing techniques.

3. **Efficiency and Scalability**: Reviewer NkHa highlights the method's efficient inference time increase of less than 3%, demonstrating the potential for practical application to existing models. The AutoCoNN toolkit mentioned by Reviewer 68he further supports the method's scalability.

4. **Expertise Incorporation and Improved Generalization**: Reviewers M9Rq and uMuv discuss the benefit of combining LMs with CoNNs, which act similarly to a Mixture of Experts (MoE), enhancing LMs' symbolic reasoning and generalization capabilities, particularly in longer-length problems.

5. **Interpretability and End-to-End Differentiability**: Reviewer uMuv values the improved interpretability, where the reasoning process is more transparent compared to black-box LM predictions, and the fact that the integration of CoNNs is end-to-end differentiable without the need for external tool APIs.

---

### Presentation:

Once again, thank you to all reviewers for your constructive comments! We have re-uploaded a new revised version, with all changes highlighted in blue, specifically:

- **The method section has been reorganized and elaborated;** *(Reviewer 68he and Reviewer M9Rq)*
- **Formula 2's notation is changed in 4.2;** *(Reviewer 68he)*
- **Corrections of some terminologies and Typos;** *(Reviewer 68he)*
- **Emphasized the performance when mixing multiple CoNN models;** *(Reviewer M9Rq and Reviewer uMuv)*
- **Added new comparison experiments on GSM8K-Hard dataset for Vanilla CoT and PAL (tool-based);** *(Reviewer NkHa, Reviewer M9Rq, and Reviewer uMuv)*
- **Added specific explanations about how to use GPT-3's API to simulate Neural Comprehension.** *(Reviewer NkHa)*
- **Modified the Supplementary Material to support uploading and downloading of CoNN through Hugging Face, thereby increasing generality and reducing cost.**  *(Reviewer 68he, and Reviewer M9Rq)*

---

### Meta-Review · Area_Chair_L8DW · 2023-12-12

**Metareview:**

This paper introduces Neural Comprehension, a novel framework that integrates Compiled Neural Networks (CoNNs) with language models to enhance their symbolic reasoning capabilities. The reviewers gave positive feedback on the submission, commending its originality, strong empirical performance, scalability, and advantages in interpretability and end-to-end differentiability. The authors have addressed reviewer concerns with additional experiments and clarifications in the revised manuscript, which include an expanded method section, notation changes, added comparative experiments, and explanations of the integration of CoNNs with GPT-3 API.

Some minor points requiring further investigation include the marginal performance improvements on certain tasks and the cost implications of CoNN development. Yet, the authors successfully demonstrate their method's efficiency and competitive performance and argue for its viability in practice over strictly tool-based approaches.

Overall, the authors have effectively responded to the reviewers' feedback, providing necessary clarifications and additional experiments that reinforce the strengths of their contributions. Given the proposed approach's significance and the experimentation depth, the paper contributes meaningfully to the domain of symbolic reasoning in language models. As such, this paper is worthy of acceptance.

**Justification For Why Not Higher Score:**

Some minor points that necessitate further investigation include the marginal performance improvements on certain tasks and the cost implications of CoNN development. Yet, the authors successfully demonstrate the efficiency and competitive performance of their method and argue for its viability in practice over strictly tool-based approaches.

**Justification For Why Not Lower Score:**

The main strengths of the paper are the innovative incorporation of rule-based computation into LMs and its efficient, scalable implementation demonstrated by less than a 3% inference time increase and the AutoCoNN toolkit. Concerns regarding generalizability and the scalability of the method across thousands of tasks have been addressed by the authors, who point out the simplicity of the gating mechanism and the potential ease of creating new CoNNs with the AutoCoNN toolkit.

---

### Decision · Program_Chairs · 2024-01-16

Accept (poster)